# WhisperSplat: Lossless Steganography in 3D Gaussian Splatting

**Nicole Meng** [1]  **Ronak Sahu** [2]  **Miao Yin** [3]  **Faysal Hossain Shezan** [3]  **Yingjie Lao** [1]

## Abstract

We present WhisperSplat, the first lossless steganography method for 3D Gaussian Splatting (3DGS) models that hides a full-resolution 2D image in a single view without any degradation of the model's rendering quality elsewhere. Prior work embeds data by retraining or modifying model weights, altering novel-view synthesis fidelity and limiting capacity. Instead, we learn a small, view-specific noise key applied to each Gaussian's spherical-harmonic (SH) features while keeping all other views remain indistinguishable from the original renders. We further propose a Gradual Pixel Perturbation (GPP) strategy with a cosine-decay schedule, bootstrapping fast divergence from the clean render before transitioning to a combined reconstruction and SSIM loss. Unlike prior works that are highly dependent on accurate and large pretrained decoders, our method is able to recover the hidden image through rendering with noise key, and an optional lightweight refiner to enhance recovery image quality. Across nine standard 3DGS data scenes, WhisperSplat demonstrates superior hidden image recovery quality without sacrifice in clean 3DGS model performance, when compared to prior work such as GS-Hider and KeySS.

## 1. Introduction

Recent advances in computer vision and graphics have enabled a wide range of applications, including film visual effects, drone mapping, and virtual reality (VR). Among emerging techniques, 3D Gaussian Splatting (3DGS) and Neural Radiance Fields (NeRF) have became leading meth-

ods for high-fidelity 3D reconstruction and scene rendering (Kerbl et al., 2023; Mildenhall et al., 2021; Meng et al., 2025; Levy et al., 2023). Owing to its superior rendering speed and visual quality, 3DGS has rapidly emerged as one of the most widely adopted 3D reconstruction frameworks (He et al., 2025). Commercial platforms such as DJI Terra and Luma AI have already integrated 3DGS into their photogrammetry and rendering pipelines, underscoring its accelerating adoption across industries (Rubloff, 2025).

While promising, the emergence of 3DGS raises significant security and privacy concerns (Zhang et al., 2025; Meng et al., 2026). 3DGS scenes are typically reconstructed from large-scale datasets or photogrammetry pipelines (Wu et al., 2024b). These assets can be easily copied, modified, or redistributed since Gaussian splats represent parameterized point clouds (Huang et al., 2024b). Moreover, adversarial manipulations (such as poisoning, merging, pruning, or compression) can subtly alter Gaussian parameters to inject malicious artifacts or erase embedded information. To address these challenges, steganography emerged as a critical technique for safeguarding 3DGS assets through watermark-based model protection (Majeed et al., 2025).

Recently, steganographic techniques tailored to 3D reconstruction models have gained increasing attention from researchers (Horváth & Józsa, 2023). Several methods have been proposed on NeRF models, including Noise-NeRF (Huang et al., 2024a), IPA-NeRF (Jiang et al., 2024), StegaNeRF (Li et al., 2023) and CopyRNeRF (Luo et al., 2023b). However, due to the distinct characteristics of 3DGS, most notably, its use of explicit Gaussian representations and splatting-based rendering (Yifan et al., 2019), as opposed to NeRF's reliance on multilayer perceptron (MLP), these existing techniques are not directly applicable. As a result, recent studies have begun to explore steganography specifically designed for 3DGS (Kerbl et al., 2023; Zhang et al., 2024).

Unlike traditional 3D pipelines, 3DGS streams full point-cloud data directly to the renderer, requiring model owners to expose all Gaussian parameters and weights for public use (Zhang et al., 2024; 2025). This transparency makes it trivial for adversaries to copy or tamper with proprietary content. Embedding imperceptible, parameter-level hidden messages during training can ensure provenance and

---

[1]Department of Electrical Engineering, Tufts University, Medford, MA, USA [2]Department of Computer Science and Engineering, University of Connecticut, Storrs, CT, USA [3]Department of Computer Science and Engineering, University of Texas at Arlington, Arlington, TX, USA. Correspondence to: Yingjie Lao <yingjie.lao@tufts.edu>.

*Proceedings of the 43 $^{rd}$ International Conference on Machine Learning*, Seoul, South Korea. PMLR 306, 2026. Copyright 2026 by the author(s).

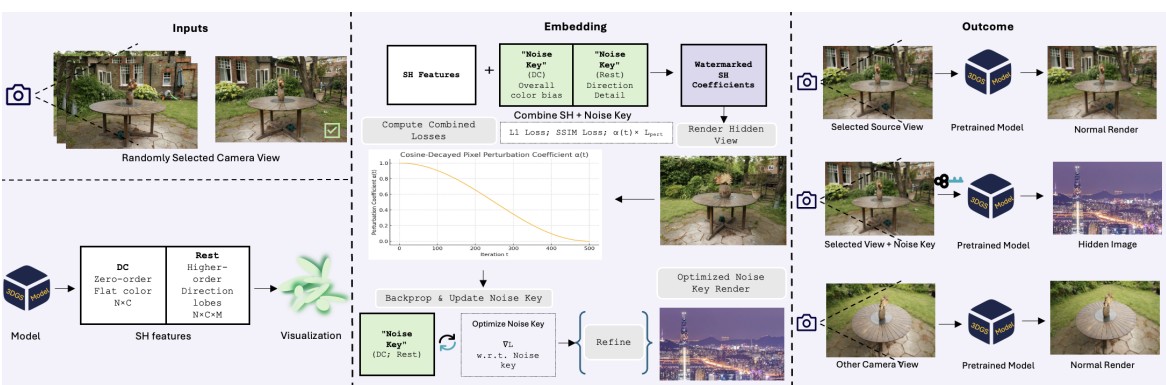

*Figure 1.* Overview of WhisperSplat's zero-footprint steganography method for 3DGS, hiding a picture of Seoul cityview. A randomly selected source view is used to extract SH feature coefficients, which are decomposed into DC (global color bias) and Rest (directional lobes). Then the noise keys are iteratively trained and added to both components. The noise keys are optimized under our cosine-decayed GPP schedule $\alpha$ with combined loss to render a hidden 2D image when a noise key is present. The "Outcome" column demonstrates the clean render when the key is not present, the stego view revealing the hidden image, and other source views remaining unchanged.

integrity without degrading rendering performance, providing essential copyright protection and secure distribution (Kerbl et al., 2023; Zhang et al., 2024; Li et al., 2023). However, as demonstrated in Table 1, current effort in 3DGS steganography have some common **limitations**:

1. All existing 3DGS steganography methods are dependent on at least one fully trained hidden image decoder for extracting hidden data, which introduces substantial computational overhead. Moreover, the steganographic capacity is inherently heavily constrained by the decoder's effectiveness.

2. These methods typically require modifying the model's weights or structure, leading to an inevitable trade-off between message capacity and clean-view fidelity. Additionally, they often require retraining the 3DGS model, which can compromise its original rendering quality when no hidden data is embedded (Zhang et al., 2024; 2025; Ren et al., 2025).

Inspired by Noise-NeRF (Huang et al., 2024a), we propose *WhisperSplat*, to the best of our knowledge, the first lossless steganography technique designed for 3DGS. The main concept is illustrated in Figure 1. WhisperSplat operates by first randomly selecting a target source view, followed by training a noise key tensor through a custom optimization process. This tensor enables the generation of the hidden image, without modifying the overall 3DGS model pipeline. Crucially, the final renderings of all other views remain consistent with the original outputs, preserving high clean-view fidelity. The secret message is revealed only when the trained noise key is applied to the selected view. Our **main contributions** can be summarized as follows:

- **First lossless 3DGS steganography:** We introduce the first steganographic method for 3DGS that embeds

hidden visual content without modifying the underlying model architecture or weights. Unlike prior approaches that require retraining or re-architecting the 3D scene, our technique maintains the original model intact, representing a novel direction in lossless 3DGS steganography.

- **Decoder-free and secure message retrieval:** Our method enables the extraction of hidden messages using only a learned noise key tensor applied to a single designated source view, eliminating the need for a bulky external decoder and ensuring the hidden image is only retrievable when the key is present. The initial hidden render can then be post-processed with a lightweight refiner to boost recovery fidelity. This refiner is roughly **8-10× smaller** than GS-Hider's message decoder (Zhang et al., 2024), and significantly reduces computational overhead.

- **High clean-view fidelity:** WhisperSplat perturbs only the input features via a learned noise key. As a result, non-secret views match the original 3DGS renders with near zero measurable degradation ($\Delta$PSNR $\approx 0$ dB), providing strong preservation of the clean views among 3DGS steganographic methods.

- **Comprehensive empirical validation:** We benchmark across all standard 3DGS scenes (e.g., bicycle, bonsai, counter), demonstrating that our method outperforms existing methods in both hidden image recovery quality and clean-view fidelity.

## 2. Related Works

3D Gaussian Splatting is becoming one of the most popular models in 3D scene reconstruction and novel view synthesis (Zhao et al., 2024; Kerbl et al., 2023; Zhang et al., 2024; Liu

| Method | Model intact | Decoder free | Clean-view fidelity |
|---|---|---|---|
| GS-Hider (Zhang et al., 2024) | No | No | Slight drop |
| KeySS (Ren et al., 2025) | No | No | Minor drop |
| **WhisperSplat (Ours)** | **Yes✓** | **Yes✓** | **No loss** |

*Table 1.* Comparison of WhisperSplat with prior 3DGS steganography methods.

et al., 2024a). Both commercial companies and academic researchers are choosing 3DGS models over NeRF models in practical use for two main reasons:

1. 3DGS is able to provide efficient real-time rendering, even at a large scale (Liu et al., 2024b).

2. 3DGS models do not rely on any neural networks to train or render scenes, which results in less computational cost. (Kerbl et al., 2023)

These differences make 3DGS the new popular model in comparison to NeRF, but it also introduces a new set of security and privacy concerns (Zhang et al., 2025).

Over the past decades, classic steganography research has not only developed sophisticated embedding and decoding algorithms for hiding audio, images, and video within deep learning models, but has placed equal emphasis on **cover fidelity**, which ensures that the presence of the hidden content remains undetectable during clean model inference (Mou et al., 2023; Roman et al., 2024; Luo et al., 2023a). More recently, steganographic schemes for NeRF, such as IPAN-eRF (Jiang et al., 2024) and StegaNeRF (Li et al., 2023) have appeared, culminating in Noise-NeRF (Huang et al., 2024a), which achieves near-perfect invisibility. In contrast, existing 3DGS methods have not placed equal emphasis on cover fidelity, i.e., clean-view fidelity in this case, relative to embedding and decoding quality, leaving a critical gap in research.

### 2.1. Current steganographic approaches for 3DGS

GS-Hider (Zhang et al., 2024) was the first proposed steganography technique on 3DGS models, followed by KeySS (Ren et al., 2025). Both of these methods modify the model's architecture or training/rendering pipeline, and they rely on external pretrained decoders to recover the hidden content. In steganography, a message decoder is a pretrained robust architecture that is responsible for recovering the exact binary hidden image embedded in an image (Krenn, 2004; Nguyen et al., 2023; Panigrahi & Padhy, 2025). Modern methods increasingly depend on large pretrained decoders, including CNNs, transformers, or diffusion models, comprising millions of parameters (Han et al., 2025; Thorat et al., 2026). These decoders require extensive fine-tuning and introduce considerable computational overhead (Chahine & Kim, 2024; Karamanji et al., 2024;

Zhang et al., 2023), as is the case with GS-Hider and KeySS (Zhang et al., 2024; Ren et al., 2025). Motivated by model ownership verification and encrypted communication (Wang et al., 2025), these two methods are effective in information hiding, laying the groundwork in the field of 3DGS steganography. Despite their prior success in this field, neither of them were able to address the seemingly unavoidable tradeoff between hidden image fidelity and clean model rendering abilities. As a result, as fidelity and payload increase, the altered model becomes increasingly detectable, making injecting hidden images no longer a secret. WhisperSplat aims to address previous limitations by proposing a decoder-free, lossless steganographic method that achieves better hidden image recovery quality and clean-view fidelity.

## 3. Preliminaries

### 3.1. 3DGS Rendering Pipeline

A 3DGS model represents a scene explicitly by $N$ oriented Gaussians, each parameterized by its 3D center position $\boldsymbol{\mu_i} \in \mathbb{R}^3$, covariance $\boldsymbol{\Sigma}_i \in \mathbb{R}^{3\times3}$, opacity $\alpha_i \in [0, 1]$, and spherical-harmonic (SH) features $\mathbf{F}_{\text{orig}}$, and SH color coefficients $\mathbf{c}_i \in \mathbb{R}^{C \times (L+1)^2}$, where $C$ represents the channels of color (i.e. 3 for an RGB image) and $L$ represents the order of SH coefficients. Specifically, the 3D covariance matrix $\boldsymbol{\Sigma}_i$ is represented by the scaling matrix $\mathbf{S}_i$ and the rotation matrix $\mathbf{R}_i$. Therefore, we have $N$ anisotropic Gaussian representations $\mathcal{G}(x) = \{G_i(x)\}_{i=1}^N$ and the 3D Gaussian can be represented in detail, where $\mathbf{x}$ represents the position of the query point:

$$G_i(\mathbf{x} : \boldsymbol{\mu}_i, \boldsymbol{\Sigma}_i) = \exp\left(-\frac{1}{2}(\mathbf{x} - \boldsymbol{\mu}_i)^\top \boldsymbol{\Sigma}_i^{-1}(\mathbf{x} - \boldsymbol{\mu}_i)\right) \tag{1}$$

where the covariance matrix $\boldsymbol{\Sigma_i}$ can be expressed as:

$$\boldsymbol{\Sigma}_i = \mathbf{R}_i\mathbf{S}_i\mathbf{S}_i^\top\mathbf{R}_i^\top \tag{2}$$

Next, each Gaussian needs to be projected from 3D to 2D during the standard rendering process. In order to complete the dimension reduction mapping of each Gaussian, the 3D center $\boldsymbol{\mu}_i$ and covariance $\boldsymbol{\Sigma}_i$ are projected onto the image plane using a combination of camera projection transformation, viewing transformation, and Jacobian of affine approximation of the projection transformation. Specifically, the 2D mean $\hat{\boldsymbol{\mu}}_i$ and 2D covariance $\hat{\boldsymbol{\Sigma}}_i$ of the projected Gaussian are computed as:

$$\hat{\boldsymbol{\mu}}_i = \mathbf{P}_i\mathbf{V}_i\boldsymbol{\mu}_i, \quad \hat{\boldsymbol{\Sigma}}_i = \mathbf{J}_i\mathbf{V}_i\boldsymbol{\Sigma}_i\mathbf{V}_i^\top\mathbf{J}_i^\top \tag{3}$$

where $\mathbf{P}_i$ is the camera projection matrix, $\mathbf{V}_i$ is the viewing transformation matrix, and $\mathbf{J}_i$ is the Jacobian of affine transformation of $\mathbf{P}_i$ evaluated at $\boldsymbol{\mu}_i$. Let $\hat{C} \in \mathbb{R}^{H\times W\times 3}$ be the $W \times H$ 2D rendered RGB by 3DGS, where $H$ is the

height of the image and $W$ is the width of the image. Then the color of the pixel at $(x, y)$ of the image is computed by alpha compositing (Kerbl et al., 2023) as:

$$\hat{C}[x, y] = \sum_{i=1}^{N} \mathbf{c}_i \sigma_i \prod_{j=1}^{i-1} (1 - \sigma_j), \qquad (4)$$

where $\sigma_i = \alpha_i G_i((x, y) : \hat{\mu}_i, \hat{\sigma}_i)$.

3DGS is fundamentally different from NeRF models due to its unique explicit scene representation and real-time rendering capabilities. 3DGS does not adopt any MLP for scene rendering. Instead, it relies on traditional computer vision techniques for representation and rendering, since using an MLP would significantly affect rendering speed (Kerbl et al., 2023). This makes the security and privacy landscape of 3DGS far different from that of NeRF, motivating our study for a unique lossless steganography method for 3DGS (Chu et al., 2026).

### 3.2. Threat Model

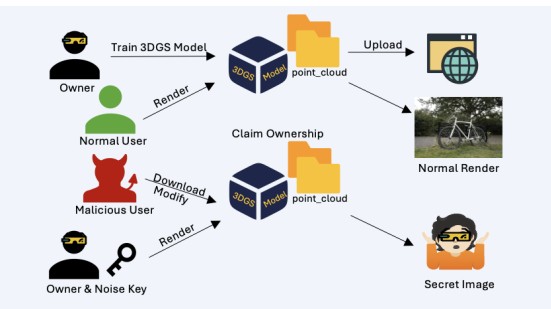

*Figure 2.* Threat Model Scenario. In the top row, we have the rightful owner, wearing distinctive yellow glasses (used as the secret image here as an example), of the 3DGS model that has invested significant time and resources into training the model before making it publicly accessible for rendering. As demonstrated in the second row, if a malicious user downloads the model, modifies it, and falsely claims ownership, the original creator can reclaim authorship using WhisperSplat. By embedding the unique noise key during training time and using it to query the model, the model is triggered to render a hidden secret image of the owner, again featuring the yellow glasses, thereby proving provenance.

As shown in Figure 2, we consider a threat scenario where a malicious actor downloads a published 3DGS model, modifies it (e.g., alters geometry, color attributes, or structure), and republishes the altered model under their own identity. Unlike implicit models (e.g., NeRF), 3DGS does not obscure internal representations, making such **tampering feasible and hard to detect**. This situation poses significant challenges for copyright enforcement and provenance verification. We assume the adversary to have full access to the published 3DGS model file, including point positions, SH coefficients, and opacity values. However, the original rendering key used by the legitimate owner remains private.

The goal of WhisperSplat is to enable secure, decoder-free verification of ownership through a stealthy steganographic mechanism. The secret message remains imperceptible in all views unless the original key tensor is applied during rendering from a specific source view.

## 4. Method

In this section, we present the detailed method of Whisper-Splat, inspired by prior steganography techniques designed for NeRF and draws upon insights from targeted adversarial attacks in computer vision (Huang et al., 2024a; Meng et al., 2024; Madry et al., 2017; Jiang et al., 2024), adapting them to the unique structure and rendering pipeline of 3DGS. As shown in Figure 1, WhisperSplat embeds a secret message by training a noise key tensor for a randomly selected source view, enabling hidden content generation without altering the 3DGS model. To render the secret image, WhisperSplat relies on the pre-trained noise key, ensuring that only individuals with access to this key can reveal the embedded content, thereby enhancing security and preventing unauthorized access or tampering. To compensate for any loss in image recovery fidelity compared to methods that use large well-trained decoders, we use a lightweight leakage refiner ($\approx$ 50–60K parameters total for the leakage coefficient map estimator plus residual predictor), which is roughly 8–10$\times$ smaller than GS-Hider's message decoder ($\approx$ 465K parameters) (Zhang et al., 2024), preserving fidelity while simultaneously reducing overhead. As opposed to prior works (Zhang et al., 2024; Ren et al., 2025) that focus on maximizing hidden image fidelity and capacity, Whisper-Splat shifts the emphasis to a key-secured pipeline with no clean-view degradation: it embeds via a learned noise key tensor for a single source view and uses a compact per-scene refiner to clean the hidden render, enabling high-quality extraction only with possession of the secret key.

### 4.1. WhisperSplat Objective

We optimize a trainable noise tensor $\delta^*$ so that a single rendering from the selected viewpoint $P^*$ reveals the hidden image $\mathbf{M}$, while without $\delta^*$ the model's outputs remain unchanged. On top of that, a lightweight refiner can be applied post hoc to suppress residual structured leakage and further clean the recovered image.

Let $\mathbf{F}_{\text{orig}}$ be the fixed SH feature representations of the pretrained 3DGS model. Let $\delta$ be the trainable noise key tensor applied to the features. We can define the perturbed features by adding the noise key as follows:

$$\mathbf{F}_{\text{pert}} = \mathbf{F}_{\text{orig}} + \boldsymbol{\delta}, \qquad (5)$$

where $\boldsymbol{\delta}$ has the same structure (e.g., dc and rest components) as the original SH features.

Let $\mathcal{R}(P; \mathbf{F})$ denote the rendering function implemented by the pretrained 3DGS model that, given a viewpoint $P$ and SH features $\mathbf{F}$, renders a 2D novel view. We then define the hidden render with key as $\mathbf{h}(\boldsymbol{\delta})$ and the clean render without key as $\mathbf{c}$, where

$$\mathbf{h}(\boldsymbol{\delta}) = \mathcal{R}(P^*; \mathbf{F}_{\text{orig}} + \boldsymbol{\delta}); \quad \mathbf{c} = \mathcal{R}(P^*; \mathbf{F}_{\text{orig}}) \quad (6)$$

We introduce the following loss components: reconstruction loss ($L_1$), structural similarity loss ($L_{ssim}$), and our push-away perturbation encouragement loss ($L_{pert}$), defined as:

$$\mathcal{L}_1(\boldsymbol{\delta}) = \big\|\mathbf{h}(\boldsymbol{\delta}) - \mathbf{M}\big\|_1, \quad (7)$$

$$\mathcal{L}_{\text{ssim}}(\boldsymbol{\delta}) = 1 - \text{SSIM}\big(\mathbf{h}(\boldsymbol{\delta}), \mathbf{M}\big), \quad (8)$$

$$\mathcal{L}_{\text{pert}}(\boldsymbol{\delta}) = -\big\|\mathbf{h}(\boldsymbol{\delta}) - \mathbf{c}\big\|_2^2. \quad (9)$$

The structural similarity loss (Wang et al., 2004) was designed to be minimized to encourage high perceptual alignment between the rendered hidden image and the hidden target. And the $L_{pert}$ encourages the rendered hidden image $h$ to deviate from the clean render early, aiming to increase embedding capacity. In order to balance between the level of our push-away perturbation encouragement loss and the reconstruction loss, we introduce the Gradual Pixel Perturbation (GPP) strategy (Loshchilov & Hutter, 2017) utilizing a cosine-decayed schedule $\alpha(t)$ that smoothly decreases from 1 to 0 over $T$ total iterations, indicated in Figure 1.

$$\alpha(t) = \frac{1}{2}\left(1 + \cos\left(\pi \frac{t}{T}\right)\right), \quad t = 1, \ldots, T \quad (10)$$

The per-step combined embedding loss is

$$\mathcal{L}(t; \boldsymbol{\delta}) = \lambda_1 \mathcal{L}_1(\boldsymbol{\delta}) + \lambda_2 \mathcal{L}_{\text{ssim}}(\boldsymbol{\delta}) + \alpha(t) \mathcal{L}_{\text{pert}}(\boldsymbol{\delta}) \quad (11)$$

with fixed scalar weights $\lambda_1 = 1$, $\lambda_2 = 0.5$. The optimal key $\boldsymbol{\delta}^*$ is obtained by iterative gradient-based minimization of this loss:

$$\boldsymbol{\delta}^{(t+1)} \leftarrow \boldsymbol{\delta}^{(t)} - \eta_t \nabla_{\boldsymbol{\delta}} \mathcal{L}(t; \boldsymbol{\delta}^{(t)}),$$

with learning rates $\eta_t$. The optimizer used in our algorithm is Adam (Kingma & Ba, 2014). After $T$ steps, the learned key $\boldsymbol{\delta}^* = \boldsymbol{\delta}^{(T)}$ produces the hidden render $\mathbf{h}(\boldsymbol{\delta}^*)$ that reveals $\mathbf{M}$ from $P^*$.

A lightweight refiner $\mathcal{F}$ can then be applied to $\mathbf{h}(\boldsymbol{\delta}^*)$ to suppress residual structured leakage and sharpen the recovered image, but it does not modify the learned noise key and is aimed to boost recovery image quality, not part of the core embedding objective. The final hidden image is then defined as $\mathbf{h}_{\text{final}} = \mathcal{F}(\mathbf{h}, \mathbf{c})$.

---

**Algorithm 1** WhisperSplat: 3DGS Steganography

**Input:** Pretrained 3DGS model with fixed SH features $\mathbf{F}_{\text{orig}}$, source viewpoint $P^*$, secret target image $\mathbf{M}$, total iterations $T$, optimizer with learning rate $\eta$, loss weights $\lambda_1, \lambda_2$, pretrained refiner $\mathcal{F}$

**Output:** optimized noise key $\boldsymbol{\delta}^*$

1: Initialize noise key $\boldsymbol{\delta} \leftarrow \mathbf{0}$
2: Compute clean render: $\mathbf{c} \leftarrow \mathcal{R}(P^*; \mathbf{F}_{\text{orig}})$
3: Initialize optimizer over $\boldsymbol{\delta}$
4: **for** $t = 1$ to $T$ **do**
5:     $\mathbf{F}_{\text{pert}} \leftarrow \mathbf{F}_{\text{orig}} + \boldsymbol{\delta}$
6:     $\mathbf{h} \leftarrow \mathcal{R}(P^*; \mathbf{F}_{\text{pert}})$
7:     $\mathcal{L}_1 \leftarrow \|\mathbf{h} - \mathbf{M}\|_1$
8:     $\mathcal{L}_{\text{ssim}} \leftarrow 1 - \text{SSIM}(\mathbf{h}, \mathbf{M})$
9:     $\mathcal{L}_{\text{pert}} \leftarrow -\|\mathbf{h} - \mathbf{c}\|_2^2$
10:    $\alpha(t) \leftarrow \frac{1}{2}(1 + \cos(\pi t/T))$    // GPP schedule
11:    $\mathcal{L} \leftarrow \lambda_1 \mathcal{L}_1 + \lambda_2 \mathcal{L}_{\text{ssim}} + \alpha(t) \mathcal{L}_{\text{pert}}$
12:    Update $\boldsymbol{\delta}$ via optimizer step on $\nabla_{\boldsymbol{\delta}} \mathcal{L}$
13: **end for**
14: $\mathbf{h}_{\text{final}} \leftarrow \mathcal{F}(\mathbf{h}, \mathbf{c})$
15: **return** optimized noise key $\boldsymbol{\delta}^*$

---

### 4.2. Algorithm

Algorithm 1 summarizes the detailed steps of WhisperSplat, which utilizes an iterative optimization process to generate noise key. In practice, we first initialize the noise key $\delta$ to be a tensor of zeros and compute the clean render from the selected source viewpoint $P^*$ to serve as a baseline. Our algorithm then uses the gradient-based optimizer Adam (Kingma & Ba, 2014) over $T$ iterations: during each step, the current key perturbs the SH features, produces a hidden render, and the combined embedding loss $\mathcal{L}$, composed of the reconstruction term, the structural similarity term, and the GPP scheduled push-away term, is computed.

After optimization, the final optimized noise key is returned, which, along with the source viewpoint $P^*$, can render the hidden image from the 3DGS model. Additionally, to compensate for the absence of a well-trained decoder like the ones in previous methods (Zhang et al., 2024; Ren et al., 2025), we can refine the final rendered hidden image through training the lightweight refiner $\mathcal{F}$ for a cleaner visualization. Concretely, we can train $\mathcal{F}$ in two stages: first estimating a smooth leakage coefficient map $\boldsymbol{\ell}$ (with ridge and total variation regularization) to remove structured contamination correlated with the clean render, and then learning a lightweight residual predictor $f$ to model the remaining discrepancy via a combination of MSE and perceptual losses, giving us the refined output.

$$\mathbf{h}_{\text{final}} = \mathcal{F}(\mathbf{h}, \mathbf{c}) = \mathbf{h} - \boldsymbol{\ell} \odot \mathbf{c} - f\big([\mathbf{c}, \mathbf{h} - \boldsymbol{\ell} \odot \mathbf{c}]\big). \quad (12)$$

This training requires access to the target $\mathbf{M}$, but at inference when we are trying to generate the hidden image with noise

key, only the learned refiner and the rendered hidden image are needed.

As demonstrated, our WhisperSplat's pipeline features a low-overhead approach: each iteration only involves the cost of a single forward render and gradient update on a small auxiliary tensor, making our method practical to run and consistent with 3DGS's real-time rendering properties. The trained noise key is compact and can be stored or transmitted easily, preserving security guarantees since only key holders can reconstruct the hidden target.

## 5. Experimental Results

To properly compare WhisperSplat with prior 3DGS steganography methods (KeySS and GS-Hider), we adopt the same pretrained model settings and dataset choices as used in prior work. We further include three scenes used in evaluating the original 3DGS model (Kerbl et al., 2023) that were not evaluated by GS-Hider and KeySS. Our evaluation focuses on two primary metrics: clean-view fidelity and hidden image recovery quality.

### 5.1. Experiment Setup

**Models** We trained the original 3DGS model (Kerbl et al., 2023) and used the checkpoints for our experiments. As both KeySS and GS-Hider provided their unmodified model rendering results as a baseline, we also provide the PSNR values of clean renderings as "3DGS Ground Truth" in our experiments Table 2.

**Datasets** Following the footsteps of 3DGS and previous steganographic methods, we conduct experiments on 9 original scenes taken from both the public Mip-NeRF360 dataset (Barron et al., 2021) and the 3DGS training dataset (Kerbl et al., 2023). On top of that, we present the results for three new scenes: train, truck, and playroom. Therefore, we are the only steganographic method that provides results on all scenes trained on the original 3DGS model.

**WhisperSplat Parameter Settings** All embedding runs use the same core configuration unless noted otherwise on Ubuntu 22.04.5 LTS (Jammy Jellyfish) with AMD EPYC 7763 64 Cores with 1 48 GB Nvidia A6000 GPU. We load a pretrained 3DGS model and select a source view (e.g., view index $P^* = 0$) for hiding the secret image. We tested different images as the secret target image used in our experiments for the paper, with the Seoul city view being one of them, and the secret images are resized to match the render resolution of the chosen view. The key optimization is performed with Adam using the following hyperparameters:

- Iterations: $T = 1000$

- Learning rate: $\eta = 1 \times 10^{-2}$

- Spherical harmonic degree: sh_degree $= 3$ (controls the capacity of the base 3DGS representation)

- View index: $v = 0$ for experimental consistency

- Loss weights: $\lambda_1 = 1.0$ for our $L_1$ loss term, $\lambda_2 = 0.5$ for the structural similarity term; the perturbation (push-away) term is cosine-decayed by $\alpha(t)$

**Performance Evaluation** As discussed prior, we focus on two main parts for performance evaluation: clean-view fidelity and image recovery quality. For both measures, we use Peak Signal-to-Noise Ratio (PSNR) values (Zhang et al., 2018; Cao et al., 2022), which measure the ratio between the maximum possible pixel intensity and the mean squared error (MSE) between two images. Higher PSNR values indicate closer agreement between two images (i.e., more accurate secret recovery) of the hidden render to the target secret image.

For clean-view fidelity, we measure how closely the post-embedding renders match the original 3DGS outputs from the same viewpoints. Specifically, we compare the PSNR values of images rendered from a source view $P$ by the original (pre-steganography) 3DGS model and the post-steganography model. Let $\mathbf{y}(P)$ and $\mathbf{y}'(P)$ denote the clean and steganographic renders, respectively. The clean-view fidelity at view $P$ is then given by

$$\mathrm{CF}(P) = \mathrm{PSNR}\big(\mathbf{y}'(P), \mathbf{y}(P)\big), \qquad (13)$$

and over a set of evaluation views $\mathcal{V}$, we report the average:

$$\mathrm{CF} = \frac{1}{|\mathcal{V}|} \sum_{P \in \mathcal{V}} \mathrm{PSNR}\big(\mathbf{y}'(P), \mathbf{y}(P)\big). \qquad (14)$$

Note that higher CF indicates stronger clean-view fidelity; ideally, CF should remain close to the "3DGS Ground Truth" values, reflecting that non-secret renders are visually indistinguishable from the original model outputs.

### 5.2. Evaluation

We report per-scene PSNR values for the recovered secret images as well as the clean render of each method in Table 2. WhisperSplat achieves the highest secret message reconstruction quality, with the highest average secret PSNR of **26.563 dB**, outperforming prior steganographic approaches such as KeySS (26.427 dB) and GS-Hider (25.178 dB), and substantially improving over retraining-style baselines like 3DGS+Decoder (21.249 dB) and 3DGS+SH (23.232 dB), introduced by GS-Hider. On individual scenes, Whisper-Splat demonstrates superior or comparable performance: for example, WhisperSplat is able to reach 26.074 dB on treehill secret image recovery vs. 22.121 dB for KeySS, 30.004 dB on garden vs. 28.179 dB, and strong recovery

*Table 2.* PSNR (dB) of clean-view and secret renders for each method and scene. 3DGS Ground Truth corresponds to the original pretrained 3DGS models and serves as the reference for clean-view fidelity. The secret-render PSNR reflects reconstruction accuracy of the hidden target image. *Since WhisperSplat does not modify any model parameters, its clean-view outputs are bit-identical to the pretrained 3DGS model; therefore its clean-view PSNR matches the 3DGS Ground Truth row by definition.*

| Methods | Scene Type | Bicycle | Bonsai | Room | Flowers | Treehill | Garden | Stump | Counter | Kitchen | Average↑ |
|---|---|---|---|---|---|---|---|---|---|---|---|
| 3DGS-GT | clean | 23.395 | 31.690 | 31.712 | 19.886 | 22.579 | 25.690 | 24.670 | 28.702 | 30.338 | 26.518 |
| 3DGS+SH | clean | 23.365 | 26.286 | 29.311 | 18.998 | 21.479 | 24.897 | 22.818 | 26.893 | 28.150 | 24.689 |
| | secret | 23.548 | 21.340 | 22.231 | 25.080 | 20.619 | 28.450 | 24.067 | 20.997 | 22.758 | 23.232 |
| 3DGS+Decoder | clean | 23.914 | 27.674 | 27.502 | 19.877 | 21.200 | 24.284 | 24.134 | 26.561 | 26.013 | 24.573 |
| | secret | 20.611 | 20.318 | 21.668 | 20.540 | 19.848 | 25.287 | 19.933 | 20.670 | 22.367 | 21.249 |
| GS-Hider (Zhang et al., 2024) | clean | 24.018 | 29.643 | 28.865 | 20.109 | 21.503 | 26.753 | 24.573 | 27.445 | 29.447 | 25.817 |
| | secret | 28.219 | 23.846 | 22.885 | 26.389 | 20.276 | 32.348 | 25.161 | 20.792 | 26.690 | 25.178 |
| KeySS (Ren et al., 2025) | clean | 23.011 | 31.081 | 30.785 | 19.476 | 22.433 | 25.225 | 23.827 | 28.120 | 29.862 | 25.980 |
| | secret | 29.533 | 25.456 | 23.877 | 29.272 | 22.121 | 28.179 | 29.452 | 20.891 | 29.064 | 26.427 |
| **WhisperSplat (Ours)** | clean | 23.395 | 31.690 | 31.712 | 19.886 | 22.579 | 25.690 | 24.670 | 28.702 | 30.338 | **26.518** |
| | secret | 29.394 | **26.254** | 23.338 | 25.250 | **26.074** | 30.004 | 27.784 | **24.203** | 26.768 | **26.563** |

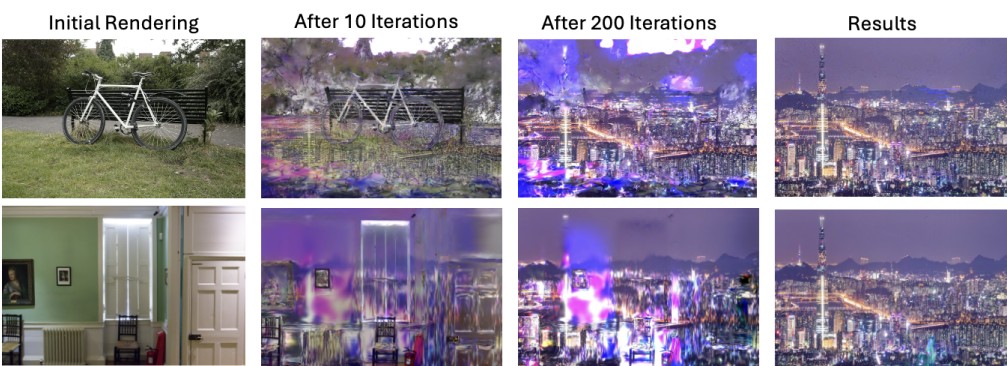

*Figure 3.* WhisperSplat's performance on example input scenes: bicycle, and drjohnson. The columns represent the optimization progress of rendered images. The initial rendering represents the clean model output when given a selected source view. After 10 iterations, we can see the effect of our GPP cosine-decay schedule: pushing pixel-level rendering away from the original renders. After 200 iterations, the combined reconstruction loss and SSIM loss start to take control, driving the rendered image towards the hidden target. After all iterations and refinement, the final renderings fully reveal the secret image with minor distortions.

on other scenes such as flowers (25.250 dB) and kitchen (26.768 dB). According to previous PSNR comparison and evaluation studies, PSNR above 25 dB is generally viewed as strong, and values within the 21.8–26 dB range indicate good visual fidelity when comparing two images (Tarigan & Isa, 2021; Keleş et al., 2021). Even in cases where another method marginally edges it on a single scene (e.g., KeySS on bicycle or stump), WhisperSplat's overall secret reconstruction profile remains the best-balanced across the full benchmark and maintains well above what is considered a good visual PSNR.

It is important to note that while our method may not achieve a significantly higher average PSNR than KeySS, this is by design. Rather than pushing for marginal gains in PSNR at the expense of increased detectability or model distortion, WhisperSplat prioritizes clean-view fidelity. In other words, our goal is to embed the secret image without introducing observable artifacts or compromising the rendering integrity of the original model. Furthermore, both KeySS and GS-

*Table 3.* PSNR (dB) of recovered secret images on additional scenes.

| Method | Playroom | Truck | Train | Avg. |
|---|---|---|---|---|
| 3DGS Ground Truth | 30.45 | 25.46 | 21.95 | 25.94 |
| WhisperSplat (secret view) | 29.12 | 26.45 | 27.80 | 27.79 |

Hider (Zhang et al., 2024; Ren et al., 2025) rely heavily on large pretrained decoders to achieve competitive image recovery fidelity. In contrast, our method requires no external decoder; it uses only a lightweight, optional refinement step post-recovery. Despite this, WhisperSplat performs comparably in secret reconstruction quality, while offering clear advantages in efficiency, robustness, and clean-view fidelity.

WhisperSplat demonstrates superior performance due to its unique embedding strategy. It maintains complete clean-view fidelity, meaning that renders on benign (non-secret) views are visually identical to the original pretrained 3DGS

outputs. Quantitatively, WhisperSplat's clean-view PSNR matches the 3DGS ground truth baseline, confirming that the embedding process introduces no measurable degradation. In contrast, GS-Hider, KeySS, 3DGS+SH, and 3DGS+Decoder all exhibit noticeable drops in clean-view PSNR, indicating partial degradation of the rendered quality and making the presence of hidden content more detectable. Unlike these prior approaches, which rely on global model modification or retraining and thus alter the cover renders, WhisperSplat fully preserves the original model's visual integrity. This combination of perfect clean-view preservation and high hidden-image fidelity establishes WhisperSplat as a robust and lossless steganographic scheme for 3DGS models.

Furthermore, we notice that previous methods do not provide performance on three scenes from the dataset for 3DGS's model evaluation (Kerbl et al., 2023): playroom, train, and truck. Therefore, we further provide WhisperSplat's performance on these scenes as shown in Table 3. It can be seen that our method maintains strong fidelity and imperceptibility across these more diverse and structurally complex environments, demonstrating WhisperSplat's broader applicability and consistency across challenging 3D settings.

We further study the sensitivity of the learning rate using using $lr = 0.1, 0.01, 0.001$ while keeping $\lambda_1 = 1$ and $\lambda_2 = 0.5$. The average PSNR values are 26.321, 26.748, and 26.648 for these learning rates. More ablation studies results on the sensitivity of loss function weights can be found in appendix.

### 5.3. Results Visualization

In Figure 3, we illustrate the effects of our GPP cosine-decay schedule loss, combined with reconstruction and SSIM loss over the course of $T = 1000$ iterations on bicycle and drjohnson. The resulting image successfully reconstructs the hidden secret image with high visual fidelity. The refinement stage further enhances detail quality, resulting in accurate and perceptually clean recovery.

### 5.4. Robustness Analysis

Following the robustness evaluation method from GS-Hider, we will assess our robustness performance via random pruning of Gaussian points and compare with GS-Hider's results. Specifically, we randomly prune a subset of Gaussian points from each watermarked model at ratios of 5%, 10%, and 25%, simulating structural degradation in the point cloud. For every pruning level, both the visible scene and hidden view are re-rendered using the same camera configuration, and PSNR/SSIM are measured relative to the unpruned baseline. Table 4 reports the degradation of the hidden view before refinement. Across scenes, *WhisperSplat* exhibits

strong resilience under moderate pruning: even with 10% of points removed, the hidden signal loses on average only $\sim$0.47 dB in PSNR and 0.05 in SSIM. At a severe pruning ratio of 25%, degradation remains bounded (average $-1.31$ dB PSNR, $-0.12$ SSIM), showing that the embedded information is spatially distributed rather than concentrated on specific anchor Gaussians. The lightweight refiner yields a minor boost in fidelity by restoring low-frequency consistency, but does not affect the underlying robustness trend. We therefore report robustness primarily in the pre-refiner stage, which directly reflects the intrinsic stability of the embedding under structural loss. We provide more robustness results in the appendix.

*Table 4.* Hidden-view degradation ($\Delta\text{PSNR}_h$, $\Delta\text{SSIM}_h$) of *WhisperSplat* before the refiner under random Gaussian pruning. Values show the loss from the unpruned baseline (ratio = 0); the last row reports the mean degradation across all scenes.

| Scene | 5% pruned | | 10% pruned | | 25% pruned | |
|---|---|---|---|---|---|---|
| | $\Delta$PSNR | $\Delta$SSIM | $\Delta$PSNR | $\Delta$SSIM | $\Delta$PSNR | $\Delta$SSIM |
| bicycle | -0.22 | -0.03 | -0.47 | -0.07 | -1.25 | -0.15 |
| bonsai | -0.24 | -0.02 | -0.37 | -0.03 | -1.05 | -0.09 |
| counter | -0.18 | -0.02 | -0.32 | -0.03 | -0.92 | -0.08 |
| drjohnson | -0.25 | -0.02 | -0.56 | -0.04 | -1.59 | -0.09 |
| flowers | -0.35 | -0.02 | -0.23 | -0.04 | -1.02 | -0.10 |
| garden | -0.30 | -0.05 | -0.59 | -0.09 | -1.78 | -0.21 |
| kitchen | -0.37 | -0.03 | -0.74 | -0.06 | -2.02 | -0.14 |
| playroom | -0.05 | -0.02 | -0.16 | -0.04 | -0.40 | -0.08 |
| room | -0.22 | -0.02 | -0.27 | -0.04 | -0.71 | -0.09 |
| stump | -0.17 | -0.03 | -0.42 | -0.06 | -1.34 | -0.15 |
| train | -0.26 | -0.03 | -0.80 | -0.05 | -2.65 | -0.13 |
| treehill | -0.11 | -0.02 | -0.09 | -0.04 | -0.43 | -0.09 |
| truck | -0.26 | -0.05 | -0.87 | -0.09 | -1.88 | -0.19 |
| **Mean** | **-0.23** | **-0.03** | **-0.47** | **-0.05** | **-1.31** | **-0.12** |

## 6. Conclusion

We introduced *WhisperSplat*, the first fully lossless steganographic framework for 3DGS models that embeds a target secret image without introducing any detectable changes to the original renderings and without relying on an external decoder. Comprehensive experiments across diverse 3DGS scenes show that WhisperSplat achieves zero measurable clean-view degradation while delivering superior secret-image recovery performance compared with GS-Hider and KeySS. Unlike previous 3DGS steganographic methods, our approach preserves all model weights intact, dispenses with any specialized trained decoder, and consistently excels in both visual fidelity and recovery accuracy.

## Acknowledgment

This work was partially supported by the National Science Foundation (NSF) under awards 2426299, 2413046, 2343618, 2532587, and 2532588.

## Impact Statement

WhisperSplat allows for imperceptible embedding of information in 3D Gaussian Splatting (3DGS) models, which raises potential concerns regarding misuse, such as unauthorized content transmission or intellectual property tampering. Although our method is designed for the embedding of secure and lossless information in legitimate applications, such as digital watermarking, the sharing of secure models, and copyright protection, we recognize the potential for adversarial use. To mitigate these risks, we encourage responsible deployment and advocate for further research into detection and safeguards against malicious steganographic practices.

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

# A. Appendix

## A.1. WhisperSplat Pipeline

We present additional technical details for our WhisperSplat method in this section, detailing our technical novelty, evaluation metric selection and calculation, and the recovery refiner pipeline.

### A.1.1. TECHNICAL DEPTH AND NOVELTY

From a high level, WhisperSplat's core optimization embedding process optimizes a small tensor *noise key*, on SH coefficients. Our idea is simple and inspired by perturbation-driven targeted adversarial attacks (Wu et al., 2024a; Sen et al., 2023). Our novelty and effectiveness come from our customized approach for the 3DGS pipeline, and our practical advantages when compared with prior decoder-based methods like GS-hider (Zhang et al., 2024) and KeySS (Ren et al., 2025). We highlight below three key technical contributions while comparing with previous methods.

**Explicit Capacity-Fidelity Control via Gradual Pixel Perturbation (GPP).** WhisperSplat introduces a gradual pixel perturbation (GPP) strategy for balancing capacity and fidelity. It is important to note that when hiding a secret message with the intention of retrieving it later, there is a balance between having enough divergence to carry the secret and being able to converge back to the target for fidelity. Prior works with large decoders can address leakage and embedding capacity issues under the learned inverse map (Zhang et al., 2024; Ren et al., 2025), but they do not explicitly introduce any scheme to manage such a balance. WhisperSplat instead, introduces the GPP cosine-decay schedule, with the push-away loss $L_{pert}$ with weight $w(t)$: we allow early iterations to drastically diverge from the clean render on a pixel level to create embedding capacity, and later iterations fade that divergence so the hidden renders align with the target image. We designed this dynamic scheduling scheme for maximum fidelity capability under limited constraints with no presence of a well-trained decoding model, highlighting our unique technical contribution.

**Standalone Noise Key with Modular Refinement.** Previous 3DGS steganographic approaches combine embedding and decoding into heavyweight learned networks, which means that the secret message retrieval is dependent on pretrained decoders and therefore involves non-trivial inference cost (Zhang et al., 2024; Ren et al., 2025; Cogranne et al., 2022). For example. KeySS's (Ren et al., 2025) embedding process involves a complete retraining of the 3DGS model at the same time as the decoder, creating large computational overhead. WhisperSplat eliminates these concerns through the implementation of the standalone noise key: the secret image is compactly stored via the optimized noise key, and presenting the key alone to the frozen pretrained 3DGS model is enough to produce the hidden render. When only the noise key is presented in the scene, even though the detailed quality of the hidden image is not ideal, the message is securely delivered and can be recognized. However, as highly detailed fidelity matters in some cases, we then introduce an optional refiner as a post-hoc cleanup tool, not part of the embedding itself. This split implementation can preserve 3DGS's real-time rendering advantage and avoid the complicated learning of an all-encompassing decoder to absorb leakage implicitly.

**Minimal Alternation on 3DGS Model for Practical Deployment** WhisperSplat does not alter the underlying 3DGS model architecture: there is no retraining of the base model, no large auxiliary network to query at inference, and no need for a joint encoder-decoder pipeline tied to specific training distributions. This makes our method extremely lightweight and portable, indicating that it is easy to integrate into existing 3DGS deployments.

### A.1.2. KEY DIFFERENCES WITH PRIOR NERF STEGANOGRAPHY METHODS

In this section, we highlight the key technical differences between our WhisperSplat method and prior steganographic methods done on NeRF models (Mildenhall et al., 2021), particularly in comparison to Noise-NeRF (Huang et al., 2024a), since we drew inspiration from this method.

1. **Location of Noise Injection:** Noise-NeRF injects noise internally, after the positional encoding step of the NeRF pipeline, modifying encoded 3D coordinates before they are processed by the network. In contrast, WhisperSplat perturbs an external noise tensor that is directly applied to the explicit Gaussian parameters (e.g., position, scale, opacity, color) used in 3D Gaussian Splatting. This perturbation occurs outside the renderer, directly modifying the rendered primitives rather than internal features of a neural network.

2. **Gradual Pixel Perturbation Strategy:** As discussed prior, WhisperSplat utilizes the GPP strategy with cosine decay for balancing capability and fidelity. Though Noise-NeRF discussed similar ideas of the push-away term, their approach is rather static, pushing pixel-level perturbation away from the original rendering only for the first 50

steps. WhisperSplat instead features a dynamic approach, slowly tuning down the weights on the push-away term and gradually shifting the importance of the reconstruction loss term and the SSIM loss term.

### A.1.3. EVALUATION METRICS

In our main paper, we presented the PSNR metric for final image recovery quality evaluation, following the evaluation metrics of previous 3DGS steganography methods (Zhang et al., 2025; 2024; Ren et al., 2025). However, SSIM values and LPIPS values are also important metrics when evaluating the quality of reconstructed scenes (Fu et al., 2023; Meng et al., 2025; Mildenhall et al., 2021). As seen in Table 5, we provide these additional measurements under the same evaluation

*Table 5.* Per-scene hidden image retrial quality measuring metrics with learning rate $lr = 0.01$, loss function weights $\lambda_1 = 1$, $\lambda_2 = 0.5$ after $T = 1000$ iterations and refiner.

| Scene | PSNR ↑ | SSIM ↑ | LPIPS ↓ |
|---|---|---|---|
| Bicycle | 29.312 | 0.891 | 0.315 |
| Bonsai | 26.250 | 0.871 | 0.289 |
| Counter | 24.224 | 0.850 | 0.332 |
| Drjohnson | 29.410 | 0.937 | 0.170 |
| Flowers | 25.216 | 0.815 | 0.354 |
| Garden | 29.941 | 0.864 | 0.277 |
| Kitchen | 26.734 | 0.907 | 0.230 |
| Room | 23.413 | 0.805 | 0.379 |
| Stump | 27.771 | 0.881 | 0.324 |
| Train | 23.312 | 0.839 | 0.271 |
| Treehill | 26.126 | 0.791 | 0.359 |
| Truck | 29.268 | 0.895 | 0.246 |

setting as our main paper's experiments. We will evaluate reconstruction quality using SSIM (Structural Similarity Index) (Wang et al., 2004), PSNR (Peak Signal-to-Noise Ratio) (Cao et al., 2022), and Learned Perceptual Image Patch Similarity (LPIPS) (Zhang et al., 2018).

SSIM measures perceptual similarity between images by comparing luminance, contrast, and structure. PSNR quantifies reconstruction quality in terms of the logarithmic ratio between the maximum possible pixel value and the mean squared error. LPIPS is a perceptual metric that measures image similarity based on deep feature representations extracted from pre-trained neural networks.

Higher values of both SSIM and PSNR indicate better visual fidelity, while lower LPIPS values indicate higher perceptual similarity. SSIM value ranges in [0,1] where 1 represents identical images. PSNR ranges between $[0, \infty]$, where the higher the number, the more identical the two images are. LPIPS ranges between [0,1] where 0 represents perfect perceptual similarity.

$$\text{PSNR} = 10 \cdot \log_{10}\left(\frac{L^2}{\text{MSE}}\right) \tag{15}$$

$$\text{MSE} = \frac{1}{MN}\sum_{i=1}^{M}\sum_{j=1}^{N}\left(I(i,j) - \hat{I}(i,j)\right)^2 \tag{16}$$

where $L$ is the maximum possible pixel value (e.g., 255 for 8-bit images), $M$ and $N$ are the image dimensions, and $I(i,j)$ and $\hat{I}(i,j)$ denote the pixel intensities of the original and reconstructed images at position $(i,j)$, respectively.

$$\text{SSIM}(x,y) = \frac{(2\mu_x\mu_y + C_1)(2\sigma_{xy} + C_2)}{(\mu_x^2 + \mu_y^2 + C_1)(\sigma_x^2 + \sigma_y^2 + C_2)} \tag{17}$$

where $\mu_x$ and $\mu_y$ are the means of images $x$ and $y$, $\sigma_x^2$ and $\sigma_y^2$ are their variances, and $\sigma_{xy}$ is the covariance. The constants $C_1 = (K_1 L)^2$ and $C_2 = (K_2 L)^2$ are used to stabilize the division, with typical values $K_1 = 0.01$, $K_2 = 0.03$, and $L = 255$.

Given two images $x$ and $y$, LPIPS computes the distance between their feature embeddings as:

$$\text{LPIPS}(x, y) = \sum_l \frac{1}{H_l W_l} \sum_{h=1}^{H_l} \sum_{w=1}^{W_l} \|w_l \odot F_l(h, w)\|_2^2 \tag{18}$$

where,

$$F_l(h, w) = f_l^x(h, w) - f_l^y(h, w) \tag{19}$$

where $f_l^x$ and $f_l^y$ are feature maps from layer $l$ of a pre-trained network (e.g., VGG or AlexNet) for images $x$ and $y$, $H_l$ and $W_l$ are the spatial dimensions of the feature map, and $w_l$ is a learned weight that calibrates the importance of each channel. For our experiments, we used VGG pre-trained network to calculate LPIPS metrics.

### A.1.4. RECOVERY REFINER

The optimizing effect of our lightweight recovery refiner that can be trained in addition to our main algorithm for a clearer hidden image recovery. We design our refiner based on the following reasoning: Given the 3DGS model and the noise key, hiding the target secret image using WhisperSplat's embedding pipeline, we have access to four pieces of information when training the refiner: the selected source camera view, the target secret image $t$, the clean rendering $c$, and the rendered hidden view $h$. Utilizing this information, we know that the structured noise seen in $h$ is coming from $c$. Therefore, we draw insights from traditional leakage removal pipelines (Roy et al., 2003; Dai et al., 2018; Feng et al., 2023) and implement a two-stage alternating refinement pipeline: first estimating a spatially smooth, patch-wise linear leakage map to subtract the dominant structured component of $c$, and then learning a nonlinear residual predictor with perceptual regularization to correct the remaining error, alternating these stages with soft blending to stabilize and progressively refine the result.

We show the mathematical details of our refiner pipeline below:

**Notation.** Let $h \in [0, 1]^{3 \times H \times W}$ denote the observed *hidden* image prior to refining, generated via noise key only, $t \in [0, 1]^{3 \times H \times W}$ the target secret image, and $c \in [0, 1]^{3 \times H \times W}$ the *clean* rendering from the base 3DGS model without the presence of noise key. All operations are element-wise or channel-wise unless specified. Define the residual between hidden and target as

$$r = h - t. \tag{20}$$

**Smooth alpha estimation.** We model the leakage as a spatially-varying linear scaling of $c$ and aim to estimate an alpha map $\alpha \in \mathbb{R}^{3 \times H \times W}$ such that the corrected hidden

$$h^{(1)} = h - \alpha \odot c \tag{21}$$

is closer to $t$. The initialization is done patch-wise: for each patch $P$ and channel, note that we allow overlapping patches as well, the local least-squares estimate is

$$\alpha_P = \frac{\langle r_P, c_P \rangle}{\langle c_P, c_P \rangle + \varepsilon}, \tag{22}$$

with $\varepsilon > 0$ a small stabilizer. These per-patch contributions are averaged in overlapping regions to produce an initial $\alpha$.

To enforce spatial smoothness and discourage overfitting, $\alpha$ is refined by solving

$$\alpha^\star = \arg\min_\alpha \frac{1}{2} \|(h - \alpha \odot c) - t\|_2^2 + \frac{\lambda_{\text{reg}}}{2} \|\alpha\|_2^2 + \lambda_{\text{tv}} \text{TV}(\alpha), \tag{23}$$

where $\text{TV}(\alpha)$ denotes the isotropic total variation (sum of absolute spatial gradients), and $\lambda_{\text{reg}}$, $\lambda_{\text{tv}}$ are regularization weights. This refinement is performed in a coarse-to-fine multi-scale sequence of patch sizes $\{P_k\}$, updating the hidden via $h \leftarrow h - \alpha^\star \odot c$ after each scale.

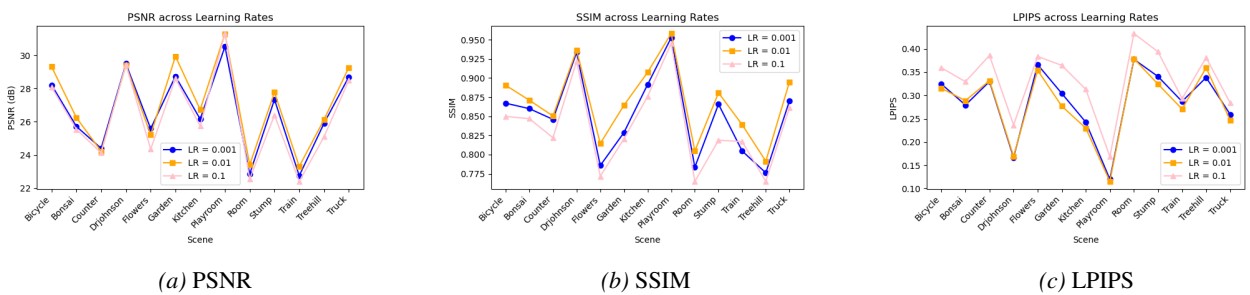

*(a)* PSNR  *(b)* SSIM  *(c)* LPIPS

*Figure 4.* Visual comparison of how different learning rate affects PSNR, SSIM, and LPIPS values across different scenes.

**Learned residual predictor.** Let $h_{\text{stage}}$ be the current hidden after alpha correction and define

$$r_{\text{stage}} = h_{\text{stage}} - t. \tag{24}$$

A convolutional network $f_\theta$ (the residual predictor) takes $c$ as input and predicts $\hat{r} = f_\theta(c)$, approximating $r_{\text{stage}}$. The corrected hidden then becomes

$$h_{\text{pred}} = h_{\text{stage}} - \hat{r}. \tag{25}$$

The predictor is trained to minimize a composite loss:

$$\mathcal{L}_{\text{pred}}(\theta) = \|\hat{r} - r_{\text{stage}}\|_2^2 + \lambda_{\text{perc}} \|\Phi(h_{\text{pred}}) - \Phi(t)\|_2^2, \tag{26}$$

where $\Phi(\cdot)$ is a perceptual feature extractor, and $\lambda_{\text{perc}}$ weights perceptual alignment.

**Alternating refinement.** After we have $h_{\text{pred}}$, the pipeline alternates between:

1. Re-estimating $\alpha$ on the updated hidden $h_{\text{pred}}$, rendering a refined hidden $h_{\text{refined}}$ with smooth alpha optimization.

2. Optionally, we can also retrain the predictor $f_\theta$ using the new residual $h_{\text{refined}} - t$, then we can produce $\hat{r}'$ and updated hidden $h'_{\text{pred}} = h_{\text{refined}} - \hat{r}'$.

**Blending step.** To stabilize transitions, intermediate corrected hidden states are blended. Given a previous hidden $h_{\text{prev}}$ and a newly corrected version $h_{\text{new}}$, we have

$$h_{\text{blend}} = \gamma\, h_{\text{new}} + (1 - \gamma)\, h_{\text{prev}}, \tag{27}$$

with $\gamma \in [0, 1]$ being the blend coefficient.

**Final output.** After the last alternation and predictor step, the final refined hidden image is

$$h_{\text{final}} = \gamma\left(h_{\text{current}} - f_\theta(c)\right) + (1 - \gamma)\, h_{\text{current}}, \tag{28}$$

where $h_{\text{current}}$ is the hidden prior to the final predictor correction.

**6. Evaluation metrics.** Metrics used by the refiner are PSNR, SSIM, and LPIPS as defined in the Section of **Evaluation Metrics**.

## A.2. Additional Experiments

We present additional experiments and more ablation study in this section, focusing on demonstrating additional metrics and convergence details for the 11 scenes on which we conducted experiments.

### A.2.1. ABLATION STUDY

In this section, we provide additional experimental results on the sensitivity of loss function weights $\lambda_1$ and $\lambda_2$ and learning rate.

### A.2.2. LEARNING RATES

We conduct additional WhisperSplat experiments with different learning rate values $lr = 0.1, 0.01, 0.001$ while loss weights are kept as $\lambda_1 = 1, \lambda_2 = 0.5$.

Tables 5, 6, and 7 present the three metrics results when learning rate $lr = 0.01, 0.001, 0.1$, respectively.

*Table 6.* Per-scene hidden image retrial quality measuring metrics with learning rate $lr = 0.001$, loss function weights $\lambda_1 = 1, \lambda_2 = 0.5$ after $T = 1000$ iterations and refiner.

| Scene | PSNR ↑ | SSIM ↑ | LPIPS ↓ |
|---|---|---|---|
| Bicycle | 28.214 | 0.867 | 0.325 |
| Bonsai | 25.722 | 0.860 | 0.279 |
| Counter | 24.403 | 0.846 | 0.330 |
| Drjohnson | 29.497 | 0.933 | 0.167 |
| Flowers | 25.587 | 0.786 | 0.366 |
| Garden | 28.741 | 0.829 | 0.304 |
| Kitchen | 26.159 | 0.891 | 0.243 |
| Playroom | 30.533 | 0.953 | 0.120 |
| Room | 22.862 | 0.784 | 0.377 |
| Stump | 27.342 | 0.866 | 0.341 |
| Train | 22.751 | 0.805 | 0.287 |
| Treehill | 25.905 | 0.776 | 0.338 |
| Truck | 28.697 | 0.870 | 0.259 |

*Table 7.* Per-scene hidden image retrial quality measuring metrics with learning rate $lr = 0.1$, loss function weights $\lambda_1 = 1, \lambda_2 = 0.5$ after $T = 1000$ iterations and refiner.

| Scene | PSNR ↑ | SSIM ↑ | LPIPS ↓ |
|---|---|---|---|
| Bicycle | 28.077 | 0.850 | 0.359 |
| Bonsai | 25.546 | 0.847 | 0.329 |
| Counter | 24.131 | 0.822 | 0.386 |
| Drjohnson | 29.448 | 0.922 | 0.237 |
| Flowers | 24.351 | 0.772 | 0.383 |
| Garden | 28.614 | 0.821 | 0.365 |
| Kitchen | 25.736 | 0.876 | 0.313 |
| Playroom | 31.315 | 0.945 | 0.169 |
| Room | 22.553 | 0.765 | 0.433 |
| Stump | 26.402 | 0.819 | 0.394 |
| Train | 22.411 | 0.817 | 0.293 |
| Treehill | 25.096 | 0.765 | 0.381 |
| Truck | 28.497 | 0.861 | 0.284 |

### A.2.3. LOSS FUNCTION WEIGHTS

As discussed in the main paper, we optimize our main loop over three loss functions, defined as:

$$\mathcal{L}_1(\boldsymbol{\delta}) = \left\| \mathbf{h}(\boldsymbol{\delta}) - \mathbf{M} \right\|_1, \tag{29}$$

$$\mathcal{L}_{\text{ssim}}(\boldsymbol{\delta}) = 1 - \text{SSIM}\big(\mathbf{h}(\boldsymbol{\delta}), \mathbf{M}\big), \tag{30}$$

$$\mathcal{L}_{\text{pert}}(\boldsymbol{\delta}) = -\left\| \mathbf{h}(\boldsymbol{\delta}) - \mathbf{c} \right\|_2^2. \tag{31}$$

We further adopt two weights $\lambda_1$ and $\lambda_2$ for the $L_1$ reconstruction loss and $L_{ssim}$, respectively:

$$\mathcal{L}(t; \boldsymbol{\delta}) = \lambda_1 \, \mathcal{L}_1(\boldsymbol{\delta}) + \lambda_2 \, \mathcal{L}_{\text{ssim}}(\boldsymbol{\delta}) + \alpha(t) \, \mathcal{L}_{\text{pert}}(\boldsymbol{\delta}) \tag{32}$$

*Table 8.* Per-scene hidden image retrial quality measuring metrics with learning rate $lr = 0.01$, loss function weights $\lambda_1 = 1, \lambda_2 = 1$ after $T = 1000$ iterations and refiner.

| Scene | PSNR ↑ | SSIM↑ | LPIPS↓ |
|---|---|---|---|
| Bicycle | 29.616 | 0.8982 | 0.306 |
| Bonsai | 26.531 | 0.8793 | 0.286 |
| Counter | 24.167 | 0.8678 | 0.308 |
| Drjohnson | 29.391 | 0.9377 | 0.167 |
| Flowers | 25.357 | 0.8168 | 0.353 |
| Garden | 30.247 | 0.8747 | 0.273 |
| Kitchen | 26.834 | 0.9108 | 0.229 |
| Playroom | 31.490 | 0.9589 | 0.112 |
| Room | 23.779 | 0.8172 | 0.370 |
| Stump | 27.996 | 0.8867 | 0.319 |
| Train | 23.315 | 0.8438 | 0.272 |
| Treehill | 26.493 | 0.8111 | 0.337 |
| Truck | 29.293 | 0.8992 | 0.241 |

*Table 9.* Per-scene hidden image retrial quality measuring metrics with learning rate $lr = 0.01$, loss function weights $\lambda_1 = 0.5, \lambda_2 = 1$ after $T = 1000$ iterations and refiner.

| Scene | PSNR ↑ | SSIM↑ | LPIPS↓ |
|---|---|---|---|
| Bicycle | 29.067 | 0.8847 | 0.288 |
| Bonsai | 26.283 | 0.8748 | 0.289 |
| Counter | 23.879 | 0.8462 | 0.337 |
| Drjohnson | 29.222 | 0.9376 | 0.166 |
| Flowers | 24.715 | 0.8085 | 0.354 |
| Garden | 29.953 | 0.8720 | 0.281 |
| Kitchen | 26.246 | 0.9074 | 0.229 |
| Playroom | 31.422 | 0.9587 | 0.121 |
| Room | 22.730 | 0.7934 | 0.398 |
| Stump | 27.401 | 0.8730 | 0.332 |
| Train | 22.985 | 0.8366 | 0.276 |
| Treehill | 25.814 | 0.8012 | 0.344 |
| Truck | 29.190 | 0.8987 | 0.242 |

In the main paper, we fixed $\lambda_1$ and $\lambda_2$ to be 1 and 0.5, respectively. That is to say, we attach more importance to the reconstruction loss than the similarity loss. In this section, we present further results with two more combinations of these weights, which are $\lambda_1 = 0.5, \lambda_2 = 1$ and $\lambda_1 = 1, \lambda_2 = 1$, respectively in Table 9 and Table 8. As we can see, the $(1, 1)$ setting provides slightly higher PSNR than our $(1, 0.5)$ choice in Table 5, whereas $(0.5, 1)$ lowers overall fidelity. We selected $\lambda_1$ and $\lambda_2$ to be 1 and 0.5 to provide the best average performance fidelity we can present.

### A.2.4. ROBUSTNESS EVALUATIONS

### A.2.5. UNAUTHORIZED EXTRACTION ATTEMPTS

We further evaluate the robustness of WhisperSplat under unauthorized extraction attempts. Specifically, we examine whether hidden content can be recovered when an attacker does not have access to the correct noise key or the designated source view. In our setting, successful recovery requires both components: the learned noise key and the source viewpoint used during embedding.

To test this, we apply randomly sampled perturbations in the SH feature space, where each perturbation has the same magnitude as the learned noise key. This experiment simulates extraction with incorrect keys, since each random perturbation corresponds to a different direction in the key space. As shown in Table 10, random perturbations consistently produce low

*Table 10.* Hidden image recovery under correct noise key and random noise key.

| Metric | bicycle | bonsai | counter | drjohnson | flowers | garden | kitchen | playroom | room | stump | train | treehill | truck |
|---|---|---|---|---|---|---|---|---|---|---|---|---|---|
| $\text{PSNR}_h$ | 9.21 | 9.99 | 8.26 | 10.58 | 9.13 | 9.76 | 9.24 | 8.37 | 8.23 | 9.64 | 10.53 | 8.74 | 9.00 |
| $\text{SSIM}_h$ | 0.170 | 0.324 | 0.231 | 0.351 | 0.178 | 0.115 | 0.236 | 0.370 | 0.240 | 0.185 | 0.340 | 0.228 | 0.138 |

*Table 11.* Hidden image recovery quality with correct key under incorrect source views.

| Metric | bicycle | bonsai | counter | drjohnson | flowers | garden | kitchen | playroom | room | stump | train | treehill | truck |
|---|---|---|---|---|---|---|---|---|---|---|---|---|---|
| $\text{PSNR}_h$ | 8.24 | 9.57 | 8.57 | 9.71 | 8.79 | 10.14 | 12.03 | 8.63 | 9.87 | 9.31 | 11.24 | 7.90 | 11.86 |
| $\text{SSIM}_h$ | 0.203 | 0.287 | 0.335 | 0.293 | 0.222 | 0.183 | 0.332 | 0.301 | 0.320 | 0.194 | 0.368 | 0.226 | 0.316 |

quality outputs across all scenes. The low $\text{PSNR}_h$ and $\text{SSIM}_h$ values indicate that the hidden image is not meaningfully recovered. These results show that the embedded content cannot be revealed by arbitrary perturbations of similar strength, and that successful extraction depends on precise alignment with the learned noise key.

We then further evaluate recovery using the correct noise key but applied at incorrect source views. As shown in Table 11, the resulting hidden image recovery quality remains consistently low across all scenes, with $\text{PSNR}_h$ and $\text{SSIM}_h$ values indicating poor alignment with the target hidden image. Despite using the correct key, applying it at non-designated viewpoints fails to produce meaningful visual structure or recognizable content. This demonstrates that the successful extraction of the hidden information requires both the correct key and the correct view.

### A.2.6. FEATURE QUANTIZATION AND SH TRUNCATION

To evaluate the robustness of WhisperSplat when Gaussian parameters are stored at reduced precision, we perform post hoc quantization of Gaussian feature parameters. Specifically, we quantize the spherical harmonics (SH) coefficients, including both the DC component and the higher order feature terms.

As shown in Table 12, under 16 bit SH feature quantization, the hidden view rendering metrics remain unchanged. Under 8 bit quantization, which is commonly used in compression and deployment settings, the degradation is negligible, with a mean $\Delta\text{PSNR}_h$ of $-0.01$ and a mean $\Delta\text{SSIM}_h$ of $-0.0034$. This suggests that hidden rendering is well preserved under realistic compression settings. When the precision is further reduced to 4 bit, hidden view performance decreases more noticeably, with a mean $\Delta\text{PSNR}_h$ of $-4.79$ and a mean $\Delta\text{SSIM}_h$ of $-0.2075$. However, we observe a similar degradation trend in the clean view as well, with a mean $\Delta\text{PSNR}_o$ of $-1.33$ and a mean $\Delta\text{SSIM}_o$ of $-0.068$. This indicates that the drop is not solely caused by degradation of the hidden signal, but also reflects the reduced representation capacity of the quantized 3DGS model.

In addition, we evaluate the robustness of WhisperSplat under SH truncation by progressively reducing the SH degree from the full setting, degree 3, to lower orders, degree 2 and degree 1. As shown in Table 12, hidden image recovery remains stable under mild truncation, with a mean $\Delta\text{PSNR}_h$ of $0.19$ and a mean $\Delta\text{SSIM}_h$ of $-0.1247$ at degree 2. As the SH degree is further reduced to degree 1, the recovery quality shows a more consistent but still moderate decrease across scenes, with a mean $\Delta\text{PSNR}_h$ of $-2.38$ and a mean $\Delta\text{SSIM}_h$ of $-0.2667$. Despite this decrease, the hidden content remains recoverable, indicating that the embedding is stable under moderate SH truncation.

*Table 12.* Hidden-view degradation of WhisperSplat under feature quantization and SH truncation. Values report the change from the full-precision, full-SH baseline. The last row reports the average degradation across all scenes.

| Scene | 16 bit Hidden | | 8 bit Hidden | | 4 bit Hidden | | Degree 2 Hidden | | Degree 1 Hidden | |
|---|---|---|---|---|---|---|---|---|---|---|
| | $\Delta\text{PSNR}_h$ | $\Delta\text{SSIM}_h$ | $\Delta\text{PSNR}_h$ | $\Delta\text{SSIM}_h$ | $\Delta\text{PSNR}_h$ | $\Delta\text{SSIM}_h$ | $\Delta\text{PSNR}_h$ | $\Delta\text{SSIM}_h$ | $\Delta\text{PSNR}_h$ | $\Delta\text{SSIM}_h$ |
| bicycle | 0.00 | 0.0000 | 0.00 | -0.0034 | -5.46 | -0.2057 | 0.10 | -0.1847 | -2.83 | -0.3668 |
| bonsai | 0.00 | 0.0000 | -0.01 | -0.0019 | -4.16 | -0.1334 | 0.55 | -0.0845 | -1.06 | -0.1795 |
| counter | 0.00 | 0.0000 | 0.00 | -0.0021 | -4.56 | -0.2642 | 1.22 | -0.1099 | -1.85 | -0.1920 |
| drjohnson | 0.00 | 0.0000 | -0.01 | -0.0017 | -3.86 | -0.1646 | -0.10 | -0.0722 | -2.88 | -0.1658 |
| flowers | 0.00 | 0.0000 | -0.01 | -0.0058 | -4.61 | -0.1548 | 0.38 | -0.1458 | -0.96 | -0.2699 |
| garden | 0.00 | 0.0000 | 0.00 | -0.0086 | -7.26 | -0.4710 | -0.37 | -0.1986 | -3.75 | -0.4461 |
| kitchen | 0.00 | 0.0000 | -0.01 | -0.0030 | -4.39 | -0.2386 | -0.87 | -0.1157 | -4.76 | -0.2949 |
| playroom | 0.00 | 0.0000 | -0.01 | -0.0033 | -3.68 | -0.1733 | 1.25 | -0.0501 | -0.16 | -0.1222 |
| room | 0.00 | 0.0000 | 0.00 | -0.0021 | -4.12 | -0.1095 | 0.44 | -0.1048 | -2.31 | -0.2195 |
| stump | 0.00 | 0.0000 | -0.01 | -0.0041 | -7.01 | -0.2302 | 0.24 | -0.1475 | -2.56 | -0.3201 |
| train | 0.00 | 0.0000 | -0.05 | -0.0008 | -7.71 | -0.1506 | -0.38 | -0.1010 | -3.62 | -0.2329 |
| treehill | 0.00 | 0.0000 | -0.01 | -0.0014 | -2.65 | -0.0960 | 0.14 | -0.1064 | -0.41 | -0.2605 |
| truck | 0.00 | 0.0000 | -0.01 | -0.0064 | -2.82 | -0.3060 | -0.12 | -0.1999 | -3.80 | -0.3899 |
| **Mean** | **0.00** | **0.0000** | **-0.01** | **-0.0034** | **-4.79** | **-0.2075** | **0.19** | **-0.1247** | **-2.38** | **-0.2667** |

