# OpenReview forum: "WhisperSplat: Lossless Steganography in 3D Gaussian Splatting"
_ICML.cc/2026/Conference — ICML 2026 regular_

### Official Review · Reviewer_ECag · 2026-03-03

**Soundness:** 3
**Presentation:** 3
**Significance:** 3
**Originality:** 2
**Overall Recommendation:** 4
**Confidence:** 3

**Summary:**

WhisperSplat presents a decoder-free steganography method for 3D Gaussian Splatting (3DGS) that learns a small view-specific noise key added to the model’s spherical-harmonic features so that one designated secret view reveals a full-resolution hidden image, while all non-secret views remain identical to the original pretrained 3DGS renders (reported as essentially zero clean-view degradation) [P1-C1], [P1-C6], [P1-C19]. The key is optimized with a combination of reconstruction and perceptual similarity losses plus a cosine-decayed “push-away” schedule that encourages early divergence from the clean render and later alignment with the target secret image [P1-C13], [P1-C33]. The paper also introduces an optional lightweight refiner to reduce leakage artifacts and improve the recovered hidden image without relying on a heavyweight decoder network [P1-C1], [P1-C15]. Experiments across standard 3DGS scenes and additional scenes report improved hidden-image recovery while maintaining perfect clean-view fidelity, and show resilience under random Gaussian pruning [P1-C19], [P1-C23].

**Compliance With Llm Reviewing Policy:**

Affirmed.

**Key Questions For Authors:**

While the threat model assumes full model access, evaluation of adversarial manipulations appears narrow (notably random Gaussian pruning); stronger tests (e.g., compression, quantization, SH truncation, view changes, or key leakage/detectability analyses) would better substantiate security and “lossless” claims.

**Strengths And Weaknesses:**

Strengths:
1. Clear presentation and solid technical soundness: The method and training procedure are described in a structured, reproducible way, with well-motivated design choices and consistent experimental reporting.
2. Lossless clean-view fidelity: The approach keeps the pretrained 3DGS model unchanged and reports that non-secret renders are bit-identical / show essentially no measurable degradation compared to the original model, improving on prior methods that introduce visible cover distortion.
3. Low-overhead, decoder-free extraction with strong evaluation: Secret recovery is achieved by rendering with a learned noise key (no heavyweight decoder at inference), with an optional small refiner for cleanup. The paper benchmarks across standard 3DGS scenes, adds additional scenes, and includes robustness testing under Gaussian pruning.

Weaknesses:
1. Originality concern relative to Noise-NeRF: The core framing, train a noise key/perturbation to reveal a target output while preserving clean behavior, is explicitly inspired by Noise-NeRF, and the main novelty is largely in adapting the idea to 3DGS plus a scheduling tweak.
2. Limited threat/robustness coverage beyond pruning: While the threat model assumes full model access, evaluation of adversarial manipulations appears narrow (notably random Gaussian pruning); stronger tests (e.g., compression, quantization, SH truncation, view changes, or key leakage/detectability analyses) would better substantiate security and “lossless” claims.

---

> ### Author Rebuttal · Authors · 2026-03-31
>
> We truly appreciate your comments and constructive review of our paper. Please kindly find our responses below.
>
> **W1.** **Originality concern relative to Noise-NeRF.**
>
> As acknowledged in the main paper, we drew our conceptual inspiration from Noise-NeRF in designing a key-conditioned trigger that reveals hidden output while preserving clean model behavior. However, WhisperSplat is not a direct adaptation. We propose a novel method specifically tailored to 3D Gaussian Splatting by formulating our distinct optimization problem for embedding and retrieval within an explicit point-based representation. We would like to note the fundamental differences between NeRF and 3DGS by highlighting the continuos volumetric fields in NeRF and the discrete Gaussian representation with spherical harmonics (SH) features in 3DGS. These differences require our method to operate under different constraints and motivate unique design choices, such as embedding in the SH feature space and a view-conditional extraction mechanism (Sections 4.1 and 4.2). In the revised version, we will clarify this distinction more explicitly and better highlight the novelty of WhisperSplat.
>
> **W2, Q1** **Limited Robustness Evaluation.**
>
> We recognize the limited robustness evaluation presented in the main paper, and present the following robustness evaluation results below.
>
> **Hidden-view degradation of WhisperSplat under feature quantization and SH truncation. Values show the loss from the full-precision, full-SH baseline; the last row reports the averaged degradation across all scenes.**
> |Scene|16-bit Hidden||8-bit Hidden||4-bit Hidden||deg-2 Hidden||deg-1 Hidden||
> |-----------|---------------|------|--------------|------|--------------|------|--------------|------|--------------|------|
> |           |$\Delta PSNR_h$|$\Delta SSIM_h$|$\Delta PSNR_h$|$\Delta SSIM_h$|$\Delta PSNR_h$|$\Delta SSIM_h$|$\Delta PSNR_h$| $\Delta SSIM_h$|$\Delta PSNR_h$|$\Delta SSIM_h$|
> | bicycle|0.00|0.0000|0.00|-0.0034|-5.46|-0.2057 | +0.10| -0.1847|-2.83|-0.3668|
> | bonsai|0.00|0.0000|-0.01|-0.0019|-4.16| -0.1334 | +0.55 | -0.0845 | -1.06 | -0.1795|
> | counter|0.00| 0.0000|0.00|-0.0021|-4.56|-0.2642|+1.22|-0.1099|-1.85|-0.1920|
> | drjohnson|0.00|0.0000|-0.01|-0.0017|-3.86|-0.1646|-0.10|-0.0722|-2.88|-0.1658|
> | flowers|0.00 |0.0000 |-0.01|-0.0058|-4.61|-0.1548|+0.38 |-0.1458|-0.96|-0.2699|
> | garden|0.00|0.0000|0.00|-0.0086|-7.26|-0.4710|-0.37|-0.1986|-3.75|-0.4461|
> | kitchen|0.00|0.0000|-0.01|-0.0030|-4.39|-0.2386|-0.87|-0.1157|-4.76|-0.2949|
> | playroom|0.00|0.0000|-0.01|-0.0033|-3.68|-0.1733|+1.25|-0.0501|-0.16|-0.1222|
> | room|0.00|0.0000|0.00|-0.0021|-4.12|-0.1095|+0.44|-0.1048|-2.31|-0.2195|
> | stump|0.00|0.0000|-0.01|-0.0041|-7.01 | -0.2302 |+0.24|-0.1475|-2.56|-0.3201|
> | train|0.00| 0.0000 |-0.05|-0.0008|-7.71|-0.1506|-0.38|-0.1010|-3.62|-0.2329|
> | treehill|0.00|0.0000|-0.01|-0.0014|-2.65|-0.0960|+0.14|-0.1064|-0.41|-0.2605|
> | truck|0.00|0.0000|-0.01|-0.0064|-2.82|-0.3060|-0.12|-0.1999|-3.80|-0.3899|
> | **Mean**|**0.00**|**0.0000**|**-0.01**|**-0.0034**|**-4.79**|**-0.2075**|**+0.19**|**-0.1247**|**-2.38**|**-0.2667**|
>
> To evaluate WhisperSplat's robustness when Gaussian parameters are stored at reduced precision, we perform a post-hoc quantization of Gaussian feature parameters. Specifically, we quantize the spherical harmonics (SH) coefficients, including both the DC component and higher-order feature terms.
>
> As shown in table above, under the 16-bit SH feature quantization, the hidden-view rendering metrics remain unchanged. Under 8-bit quantization, which is considered standard in many compression and deployment settings, the degradation is negligible (mean $\Delta PSNRₕ = -0.01$, $\Delta SSIMₕ = -0.0034$). This suggests that hidden rendering is preserved well under realistic compression settings. As the precision is further reduced to 4-bit, we can see a drop in the hidden-view performance (mean $\Delta PSNRₕ = -4.79$, $\Delta SSIMₕ = -0.21$). However, we observed the same trend in the clean-view degradation as well, with mean $\Delta PSNRₒ = -1.33$, $\Delta SSIMₒ = -0.068$. This indicates that the drop is not solely due to degradation of the hidden signal, but also reflects on the reduced representation capacity.
>
> In addition, we evaluate the robustness of WhisperSplat under SH truncation by progressively reducing the SH degree from the full setting (degree = 3) to lower orders (degree = 2 and 1). As shown in the table above, hidden image recovery quality remains largely unchanged under mild truncation (PSNR rises in some cases from smoothing under SH truncation), with negligible degradation at degree 2. As the SH degree is further reduced to 1, the recovery quality shows a consistent but still moderate decrease across scenes (mean $\Delta PSNR_h = -2.38$, $\Delta SSIM_h = -0.27$). Despite the slight drop in recovery quality, the hidden content remains clearly recoverable, indicating that the embedding is stable under moderate SH truncation.

---

> > ### Author Rebuttal · Reviewer_ECag · 2026-04-03
> >
> > Thank you! The review have resolved all my concerns.

---

> > > ### Author Response · Authors · 2026-04-05
> > >
> > > We thank the reviewer for the valuable comments and insightful questions. We appreciate your support and thoughtful engagement during the rebuttal process, which has helped us further improve the paper.

---

### Official Review · Reviewer_z1Tt · 2026-03-12

**Soundness:** 3
**Presentation:** 3
**Significance:** 4
**Originality:** 4
**Overall Recommendation:** 5
**Confidence:** 3

**Summary:**

The proposed method is the first lossless 3DGS steganographic method. Overall, a central concept studied by the manuscript is whether a frozen 3DGS asset can serve as a carrier for a hidden image that is only revealed when a learned external key is applied at inference.

**Compliance With Llm Reviewing Policy:**

Affirmed.

**Key Questions For Authors:**

Please address my comments!

**Limitations:**

Mmm... On the limitations side, the discussion is too weak: there is no serious treatment of key leakage and no real steganalysis evaluation.

**Strengths And Weaknesses:**

I like the idea - it's novel and original.

A few notes:
- "All existing 3DGS steganography methods is dependent on at least one" -> "All existing 3DGS steganography methods *are* dependent on at least one"
- I would also push the authors on what exactly “lossless” means here. The secret image recovery is obviously not lossless in the literal sense -  the paper itself talks about “minor distortions” and refinement. So “lossless” here really means the original 3DGS model has untouched views, not that the hidden image is recovered exactly. That needs to be stated much more carefully

---

> ### Author Rebuttal · Authors · 2026-03-31
>
> We thank you for taking the time to review our paper and providing your invaluable feedback. Please find our rebuttal response below.
>
> **W1.** **Sentence Grammar.**
> Thank you for poining out the grammatical error. We will be sure to fix this and perform additional proofreading for the later version.
>
> **W2.** **Clarification on "lossless".**
> We thank the reviewer for this important clarification, and we agree that the term "lossless" can be ambiguous in our current wording. In our work, "lossless" specifically refers to the fact that the original 3DGS rendering quality remains unchanged before and after the hidden information, instead of the perfect recovery of the hidden images. In other words, the embedding process does not introduce any visible degradation in the rendered scenes, and the model preserves its original functionality perfectly with the presence of hidden information.
>
> As noted in the paper, the recovered rendered hidden image might exhibit minor distortions due to the imperfect optimization process. We will be sure to revise the definition of "lossless" in the later version of the paper to explicity distinguish between these two aspects and avoid any confusion.
>
> **Q1.** **Limitation on key leakage and steganalysis.**
>
> We thank the reviewer for highlighting the importance of this discussion. In WhisperSplat, we do not modify the stored 3DGS model structure. Instead, the hidden content is only revealed when the correct noise key is applied during extraction at designated source view. As a result, conventional model-level steganalysis is less applicable in our setting, since there is no persistent change in the published 3DGS model for an attack to analyze and potentially detect our secret embedding. Under our method pipeline settings and formulations, the key leakage and security question would be the noise key exposure: without access to the correct noise key and knowing which source view to apply to, an attacker cannot directly recover the hidden content from the model alone.
>
> To further examine this, we conduct an additional experiment by applying randomly sampled perturbations in the SH feature space with the same magnitude as the noise key. As shown in the table below, these random perturbations consistently produce low-quality outputs across all scenes, with $PSNR_h$ and $SSIM_h$ values indicating no meaningful recovery of the hidden image. This experiment aims to stimulate the effect of using incorrect or random keys, since each perturbation represents a different direction in the key space. Our results demonstrate that random noise keys do not reveal the embedded image, and that successful extraction requires precise alignment with the correct key.
>
> **Hidden image recovery under correct noise key and random noise key**
> | Metric   | bicycle | bonsai | counter | drjohnson | flowers | garden | kitchen | playroom | room  | stump | train | treehill | truck |
> | -------- | ------- | ------ | ------- | --------- | ------- | ------ | ------- | -------- | ----- | ----- | ----- | -------- | ----- |
> | $PSNR_h$ | 9.21    | 9.99   | 8.26    | 10.58     | 9.13    | 9.76   | 9.24    | 8.37     | 8.23  | 9.64  | 10.53 | 8.74     | 9.00  |
> | $SSIM_h$ | 0.170   | 0.324  | 0.231   | 0.351     | 0.178   | 0.115  | 0.236   | 0.370    | 0.240 | 0.185 | 0.340 | 0.228    | 0.138 |
>
> We then further evaluate recovery using the correct noise key but applied at incorrect source views. As shown in the table below, the resulting hidden image recovery quality remains consistently low across all scenes, with $PSNR_h$ and $SSIM_h$ values indicating poor alignment with the target hidden image. Despite using the correct key, applying it at non-designated viewpoints fails to produce meaningful visual structure or recognizable content. This demonstrates that the successful extraction of the hidden information requires both the correct key and the correct view.
>
> **Hidden image recovery quality with correct key under incorrect source views.**
> | Metric   | bicycle | bonsai | counter | drjohnson | flowers | garden | kitchen | playroom | room  | stump | train | treehill | truck |
> | -------- | ------- | ------ | ------- | --------- | ------- | ------ | ------- | -------- | ----- | ----- | ----- | -------- | ----- |
> | $PSNR_h$ | 8.24    | 9.57   | 8.57    | 9.71      | 8.79    | 10.14  | 12.03   | 8.63     | 9.87  | 9.31  | 11.24 | 7.90     | 11.86 |
> | $SSIM_h$ | 0.203   | 0.287  | 0.335   | 0.293     | 0.222   | 0.183  | 0.332   | 0.301    | 0.320 | 0.194 | 0.368 | 0.226    | 0.316 |
>
> We will add these analysis and more discussions in the later version.

---

> > ### Author Rebuttal · Reviewer_z1Tt · 2026-03-31
> >
> > Thank you for the clarifications. I have updated my score.

---

> > > ### Author Response · Authors · 2026-04-05
> > >
> > > Thank you for the update and for taking the time to read our rebuttal. We appreciate the thoughtful review and your helpful suggestions!

---

### Official Review · Reviewer_gGBZ · 2026-03-12

**Soundness:** 3
**Presentation:** 3
**Significance:** 3
**Originality:** 3
**Overall Recommendation:** 4
**Confidence:** 3

**Summary:**

This paper presents WhisperSplat, a method aimed at improving the storage efficiency and representation of 3D Gaussian Splatting (3DGS) models. The authors propose a compression-oriented framework designed to reduce memory footprint while preserving rendering quality. The approach focuses on compact encoding and efficient storage of Gaussian parameters, with the goal of enabling more scalable deployment of Gaussian-based scene representations.

Experimental results show that the proposed method significantly reduces storage size while maintaining comparable rendering performance to the original 3DGS representation.

**Compliance With Llm Reviewing Policy:**

Affirmed.

**Key Questions For Authors:**

1. How sensitive is the compression performance to scene complexity or Gaussian count?
2. Does the proposed method introduce additional computational overhead during rendering?
3. How does the method perform on very large scenes compared with other compression approaches?
4. Is the compression loss truly negligible in all tested scenarios, or are there cases where artifacts become noticeable?

**Limitations:**

- The method focuses primarily on storage reduction and does not significantly change the underlying rendering model.
- The scalability to extremely large scenes or long sequences is not fully explored.

**Strengths And Weaknesses:**

## Strengths
- The paper addresses a practical and increasingly important problem in the Gaussian splatting ecosystem: the large storage footprint of trained 3DGS models.
- The proposed approach is conceptually straightforward and appears easy to integrate into existing Gaussian pipelines.
- Experimental results demonstrate meaningful reductions in model size while maintaining similar rendering quality.
- The work has potential relevance for real-world applications such as large-scale scene deployment, mobile rendering, or web-based visualization.

## Weaknesses
- The novelty of the approach appears somewhat incremental, as several recent works have also explored compression or efficient representations for Gaussian splatting.
- The paper could provide a more thorough comparison with related compression or model reduction techniques.
- Some implementation details are not fully described, making it slightly difficult to evaluate reproducibility.
- The impact of compression on more challenging scenes (e.g., highly detailed geometry or complex lighting) could be further analyzed.

---

> ### Author Rebuttal · Authors · 2026-03-31
>
> We really appreciate the review, and please find our responses below.
>
> **W1, W2, W4.** **Novelty of WhisperSplat, more comparison, and its impact.**
> We would like to respectfully clarify that WhisperSplat is not a compression or model reduction method. Instead, its primary goal is steganographic embedding, enabling high-fidelity information to be hidden within a 3DGS model without affecting rendering quality.
>
> Unlike compression methods, which trade off fidelity for storage efficiency, WhisperSplat maintains the original model behavior while hiding a full resolution 2D image within the model. Our novelty lies within the fact that we do not sacriface original model rendering quality, marking us, to the best of our knowledge, the first 3DGS steganographic method that is lossless.
>
> **W3.** **Implementation Details.**
> We thank the reviewer for pointing this out. We note that the complete pipeline implementation and environment setup were provided with the submission in WhisperSplat.zip, including all key components such as noise key optimization, hidden image extraction, and thorough evaluation. We will publish our code base along with the camera-ready version to ensure the reproducibility of WhisperSplat. To address the reviewer's concern, we will make sure that the paper more explicitly describes implementation details of our key components to match the released code.
>
>
> **Q1.** **Sensitivity to scene complexity or Gaussian count.**
> WhisperSplat was evaluated across multiple benchmark scenes where each scene has different Gaussian counts. The breakdown of our per scene performance is shown in Table 2 of the main paper, and we observe stable hidden image recovery and rendering quality.
>
> From a design perspective, the embedding is applied to the SH features of the 3DGS representations, and is distributed across active Gaussians. As demonstrated via our random pruning robustness evaluation in Section 5.4, the embedding is not highly dependent on any particular small subset of primitives. Across scenes,WhisperSplat exhibits strong resilience under moderate pruning: even with 10%  of points removed, the hidden signal loses on average only ∼0.47dB in PSNR and 0.05 in SSIM. Therefore, our distributed embedding indicates that our method is not inherently sensitive to a specific Gaussian count.
>
> **Q2.** **Computational overhead during rendering.**
> By design, WhisperSplat introduces no additional computational overhead during standard 3DGS rendering. In our implementation, the learned noise key is only used during the hidden image extration process on the designated pre-selected source view. During normal rendering, the Gaussian parameters are used as in the originaly 3DGS pipeline without additional computation needed.
>
> During the hidden image extraction process, since the noise key has already been optimized and trained, adding the key at the designated source view can be directly fed to the 3DGS model pipeline to recover the hidden content without requiring further training or optimization of the model. Therefore, our method will not introduce further computational costs or affect normal rendering speed.
>
> **Q3.** **Performance on large scenes.**
> As clarified before, WhisperSplat does not claim to compress the model or introduce any new structural components, our method operates directly on exisiting 3DGS representations. Therefore, the scalability of WhipserSplat follows that of standard 3DGS models. The embedding and extraction pipeline does not alter with increasing scene size or model capability.
>
> **Q4.** **True lossless and artifacts.**
> Since WhisperSplat does not alter model structure nor affect normal model performance when noise key is not present, we observe that WhisperSplat preserves the visual quality of all normally rendered scenes by the 3DGS model after hidden image embedding. We further clarify that the "minor distortions and artifacts" mentioned in the paper refers to the recovered hidden image quality, rather than the original rendered scene. As demonstrated in our experimental results table, standard rendered views from the 3DGS model remain visually indistinguishable from the original model, measured in PSNR degradation of 0dB, as shown in Table 2 in our main paper.

---

> > ### Author Rebuttal · Reviewer_gGBZ · 2026-04-07
> >
> > Thanks for the authors' response. I will keep my rating.

---

### Official Review · Reviewer_Y7XR · 2026-03-13

**Soundness:** 2
**Presentation:** 3
**Significance:** 3
**Originality:** 3
**Overall Recommendation:** 3
**Confidence:** 4

**Summary:**

This paper presents WhisperSplat, a lossless steganography framework for 3D Gaussian Splatting (3DGS) models. Unlike prior methods that modify model parameters or rely on external decoders, WhisperSplat embeds a secret 2D image by optimizing a compact noise key tensor applied to the spherical harmonic features of a frozen, pre-trained 3DGS model. The hidden image is revealed only when the key is used during rendering from a specific source viewpoint, while all other views remain bit-identical to the original, ensuring perfect stealth.

**Compliance With Llm Reviewing Policy:**

Affirmed.

**Final Justification:**

The last reply from the authors stated "Given that training a 3DGS requires a significant amount of time and computational cost, it is crucial to protect the copyright, integrity, and privacy of such 3D assets. Steganography, as a crucial technique for encrypted transmission and copyright protection, has been extensively studied. However, it still lacks profound exploration targeted at 3DGS."

The focus seems still on copyright protection etc., from which the reviewer think the proposed method cannot provide sufficient copyright protection.

The authors also cite several prior related works, but did not really explain the benefits to hide an image in a 3DGS asset for secure transmission of an 2D image. There are many more practical ways to do so.

In summary, I am still not fully being convinced by the motivation of the work. Therefore, I would keep my original rating.

**Key Questions For Authors:**

n.a.

**Limitations:**

n.a.

**Strengths And Weaknesses:**

## Strengths
1. The central concept is both clear and relatively novel within the context of 3DGS steganography: rather than retraining the model or embedding information directly into its parameters, the approach preserves the original 3DGS representation entirely and instead learns an external key tensor.

2. while the method is simple and easy to follow, the empirical results are reasonablely good.

## Weaknesses
1. The evaluations on the robustness is insufficient. The evaluations should include more than Gaussian prunning. How about the robustness against Gaussian quantization, fine-tuning, SH truncation etc.?

2. What if the attacker train their own key at the same/different viewpoint using a different hidden image? Both the original owner and the attacher can claim the ownership. How the method could avoid this kind of situation?

Overall, the reviewer think the idea is simple but effective. The main concern lies at the additional security concerns and robustness of the method. If the authors could provide reasonable rebuttal, the reviewer could consider improve his rating.

---

> ### Author Rebuttal · Authors · 2026-03-31
>
> We sincerely appreciate your feedback and suggestions, please find our responses below.
>
> **W1.** **Robustness evaluation.**
> In our submission, we focused on random pruning to evaluate robustness (Table 4), which is consistent with prior 3DGS steganography work such as GS-Hider for fair comparison [1]. It provides a controlled way to test whether the hidden signal depends on a subset of Gaussians. Below, we provide additional robustness tests and will incorporate a more comprehensive evaluation in the later version.
>
> **Hidden-view degradation of WhisperSplat under feature quantization and SH truncation. Values show the loss from the full-precision, full-SH baseline**
> |Scene|16-bit||8-bit||4-bit||deg-2||deg-1||
> |-----------|---------------|------|--------------|------|--------------|------|--------------|------|--------------|------|
> |           |$\Delta PSNR_h$|$\Delta SSIM_h$|$\Delta PSNR_h$|$\Delta SSIM_h$|$\Delta PSNR_h$|$\Delta SSIM_h$|$\Delta PSNR_h$| $\Delta SSIM_h$|$\Delta PSNR_h$|$\Delta SSIM_h$|
> | bicycle|0.00|0.0000|0.00|-0.0034|-5.46|-0.2057 | +0.10| -0.1847|-2.83|-0.3668|
> | bonsai|0.00|0.0000|-0.01|-0.0019|-4.16| -0.1334 | +0.55 | -0.0845 | -1.06 | -0.1795|
> | counter|0.00| 0.0000|0.00|-0.0021|-4.56|-0.2642|+1.22|-0.1099|-1.85|-0.1920|
> | drjohnson|0.00|0.0000|-0.01|-0.0017|-3.86|-0.1646|-0.10|-0.0722|-2.88|-0.1658|
> | flowers|0.00 |0.0000 |-0.01|-0.0058|-4.61|-0.1548|+0.38 |-0.1458|-0.96|-0.2699|
> | garden|0.00|0.0000|0.00|-0.0086|-7.26|-0.4710|-0.37|-0.1986|-3.75|-0.4461|
> | kitchen|0.00|0.0000|-0.01|-0.0030|-4.39|-0.2386|-0.87|-0.1157|-4.76|-0.2949|
> | playroom|0.00|0.0000|-0.01|-0.0033|-3.68|-0.1733|+1.25|-0.0501|-0.16|-0.1222|
> | room|0.00|0.0000|0.00|-0.0021|-4.12|-0.1095|+0.44|-0.1048|-2.31|-0.2195|
> | stump|0.00|0.0000|-0.01|-0.0041|-7.01 | -0.2302 |+0.24|-0.1475|-2.56|-0.3201|
> | train|0.00| 0.0000 |-0.05|-0.0008|-7.71|-0.1506|-0.38|-0.1010|-3.62|-0.2329|
> | treehill|0.00|0.0000|-0.01|-0.0014|-2.65|-0.0960|+0.14|-0.1064|-0.41|-0.2605|
> | truck|0.00|0.0000|-0.01|-0.0064|-2.82|-0.3060|-0.12|-0.1999|-3.80|-0.3899|
> | **Mean**|**0.00**|**0.0000**|**-0.01**|**-0.0034**|**-4.79**|**-0.2075**|**+0.19**|**-0.1247**|**-2.38**|**-0.2667**|
>
> As shown in table above, under 16-bit SH feature quantization, the hidden-view rendering metrics remain unchanged. Under 8-bit quantization, which is standard in many compression and deployment settings [3, 4], the degradation is negligible. As precision is further reduced to 4-bit, we can see a drop in hidden-view performance. However, we observed the same trend in clean-view degradation as well, which indicates that the drop is not solely due to degradation of the hidden signal, but also the reduced representation capacity. In addition, we evaluate robustness under SH truncation by reducing the SH degree from the full setting to lower orders. As shown in the table, hidden image recovery remains largely unchanged under mild truncation (PSNR rises in some cases from smoothing under SH truncation), with negligible degradation at degree 2. As the degree is reduced to 1, recovery quality shows a consistent but still moderate decrease across scenes.
>
> **W2.** **New key generation.**
> Thanks for the insightful comment. As our title suggests, we consider the main application of WhisperSplat to be steganography, where the goal is to conceal messages within a 3DGS model for encrypted communication, secure information sharing, and digital watermarking. In our method design, we prioritize hidden information concealment, where imperceptibility and non-destructive embedding are the primary objectives. In our evaluations, we compared with prior steganographic techniques [1, 2] on hidden image recovery quality and model performance degradation.
>
> We agree that ownership verification may only be used when the attacker is non-adaptive. This is expected, since WhisperSplat serves as a lossless steganographic method for 3DGS models that allows model owners to hide a full-resolution 2D image within the 3D model without altering performance. Moreover, since WhisperSplat does not modify model behavior and the secret image can only be recovered with the noise key, the method is highly stealthy, making it unlikely for an attacker to detect embedding.
>
> We also note that even if an attacker trains a new key to embed their own information, the resulting model would retain both the original owner's key and the attacker's key as valid. This does not contradict that the original owner can still present their key and recover the original hidden content, achieving successful steganography. We will further clarify the scope of our work in the later version.
>
> **References**
>
> [1] Gs-hider: Hiding messages into 3D Gaussian splatting, NeurIPS, 2024.
>
> [2] Key-secured 3D secrets within 3D Gaussian splatting, arXiv, 2025.
>
> [3] Quantization for efficient integer-only inference, CVPR, 2018.
>
> [4] Neural network quantization: A white paper, arXiv, 2021.
>
> [5] Universally quantized neural compression, NeurIPS, 2020.

---

> > ### Author Rebuttal · Reviewer_Y7XR · 2026-04-03
> >
> > From the introduction, the paper emphasizes the main purposes are watermark protection, copyright protection, secure distribution, from this perspective, I do not think the proposed method is effective as what I said in Weakness #2. The current method could not achieve the desired purpose.
> >
> > If the purpose is simply to hide an image in a 3DGS asset as the author replied in the rebuttal, I am wondering if it is really necessary to do so from a practical usage scenario. There are many ways to hide an image or some important information. What would be the benefits to do so?

---

> > > ### Author Response · Authors · 2026-04-05
> > >
> > > We sincerely thank the reviewer for the follow-up questions. WhisperSplat is not positioned as a standalone copyright enforcement mechanism, but rather as a steganographic technique for 3D Gaussian Splatting (3DGS) models. To better clarify the scope of steganography, we refer to the motivation and application of steganography from GS-Hider [1]: "Given that training a 3DGS requires a significant amount of time and computational cost, it is crucial to protect the copyright, integrity, and privacy of such 3D assets. Steganography, as a crucial technique for encrypted transmission and copyright protection, has been extensively studied. However, it still lacks profound exploration targeted at 3DGS." Steganography has been extensively studied across a range of machine learning architectures. Recently, significant research interest has also arisen in the context of 3D reconstruction. To provide more context, we provide a brief review of some representative recent works of steganography on both Neural Radiance Fields (NeRF) models and 3DGS models below.
> > >
> > > The first steganographic method on NeRF models was introduced in 2023, where the model generates a secret viewpoint image during rendering for secure message delivery purposes [2]. Since then, several follow-up works have been proposed. For instance, Stega-NeRF [3] embeds imperceptible information into NeRF models, with ownership verification as a primary application. Stega4NeRF [4] trains a NeRF model seletcing viewpoint in 3D space to generate unique secret viewpoint image for the purpose of "securely transmitting the message", while Noise-NeRF [5] addresses preivous retraining limitations of steganographic methods.
> > >
> > > With the invention of 3DGS, which achieves real-time photo-realistic rendering of 3D scenes, researchers have shifted their attention more on 3DGS. The field of 3DGS steganography naturally became a topic of interest. Last year, KeySS [6] proposed a novel key-secured 3D steganography framework for "secure digital commuincations" and "copyright protection", which was recently published in ICLR2026. GS-Hider [1] then introduced a dual-decoder framework where a single 3DGS model supports public rendering while allowing only authorized users to extract hidden content, highlighting the importance of binding information 3D representations. As illustrated in Figure 1 of GS-Hider, the threat model assumes that the 3DGS asset is publicly distributed and can be freely rendered by any user, whereas only authorized parties equipped with a decoder can access the embedded information. This aligns with our assumption that model access is open, while extraction remains controlled. Different from these prior works, our work presents the first lossless steganography method for 3DGS. We also note that we have compared WhisperSplat's performance with both KeySS and GS-Hider in Table 4 of the main paper. We understand the confusion from the reviewer on the importance of the field of 3D steganography, and we will include further clarifications in our later version.
> > >
> > > **References:**
> > >
> > > [1] Zhang et al., *GS-Hider: Hiding Messages into 3D Gaussian Splatting*, NeurIPS, 2024.
> > >
> > > [2] Dong et al., *Steganography for Neural Radiance Fields by Backdooring*, arXiv, 2023.
> > >
> > > [3] Li et al., *StegaNeRF: Embedding Invisible Information within Neural Radiance Fields*, ICCV, 2023.
> > >
> > > [4] Dong et al., *Stega4NeRF: Cover Selection Steganography for Neural Radiance Fields*, J. Electronic Imaging, 2024.
> > >
> > > [5] Huang et al., *Noise-NeRF: Hide Information in Neural Radiance Field Using Trainable Noise*, ICANN, 2024.
> > >
> > > [6] Ren et al., *All That Glitters Is Not Gold: Key-Secured 3D Secrets within 3D Gaussian Splatting*, arXiv, 2025.

---

### Decision · Program_Chairs · 2026-04-30

**Decision:**

Accept (regular)

**Comment:**

The paper presents WhisperSplat, a steganography framework for 3D Gaussian Splatting that embeds a 2D image into a single view while preserving lossless novel-view synthesis. Reviewer scores lean positive, with three positive recommendations (two Weak Accepts and one Accept) and one Weak Reject. Reviewers generally find the method technically sound and appreciate the strict “lossless” property, namely that non-secret views remain bit-identical to the original renders. The main concerns focus on limited robustness evaluation, especially under quantization and truncation, and on whether the method's practical scope is better understood as steganography rather than as stronger copyright or ownership protection. The rebuttal adds useful robustness experiments and clarifies the method's intended scope, which addresses most of these concerns and leaves the overall discussion positive. The AC supports acceptance.